# Receptor-driven, multimodal mapping of cortical areas in the macaque monkey intraparietal sulcus

Meiqi Niu[1], Daniele Impieri[1], Lucija Rapan[1], Thomas Funck[1], Nicola Palomero-Gallagher[1,2,3,4]*, Karl Zilles[1,4]

[1]Institute of Neuroscience and Medicine (INM-1), Research Centre Jülich, Jülich, Germany; [2]Department of Psychiatry, Psychotherapy and Psychosomatics, Medical Faculty, Aachen, Germany; [3]C. & O. Vogt Institute for Brain Research, Heinrich-Heine-University, Düsseldorf, Germany; [4]JARA-BRAIN, Jülich-Aachen Research Alliance, Jülich, Germany

**Abstract** The intraparietal sulcus (IPS) is structurally and functionally heterogeneous. We performed a quantitative cyto-/myelo- and receptor architectonical analysis to provide a multimodal map of the macaque IPS. We identified 17 cortical areas, including novel areas PEipe, PEipi (external and internal subdivisions of PEip), and MIPd. Multivariate analyses of receptor densities resulted in a grouping of areas based on the degree of (dis)similarity of their receptor architecture: a cluster encompassing areas located in the posterior portion of the IPS and associated mainly with the processing of visual information, a cluster including areas found in the anterior portion of the IPS and involved in sensorimotor processing, and an 'intermediate' cluster of multimodal association areas. Thus, differences in cyto-/myelo- and receptor architecture segregate the cortical ribbon within the IPS, and receptor fingerprints provide novel insights into the relationship between the structural and functional segregation of this brain region in the macaque monkey.

*For correspondence:
n.palomero-gallagher@fz-juelich.de

Competing interests: The authors declare that no competing interests exist.

## Introduction

In primates, the intraparietal sulcus (IPS) serves as an interface between the visual and sensorimotor systems in order to integrate visual and somatosensory modalities. Thus, it represents a region involved in multiple functions. Besides planning and execution of movements and eye movements (*Jeannerod et al., 1994*; *Sakata et al., 1997*; *Seo et al., 2009*), it is also involved in different aspects of sensorimotor integration, and also in spatial attention (*Andersen et al., 1985*; *Andersen, 1995*; *Colby and Duhamel, 1991*; *Duhamel et al., 1998*; *Grefkes and Fink, 2005*; *Grefkes et al., 2002*; *Bremmer et al., 2001*; *Pessoa et al., 2003*; *Hadjidimitrakis et al., 2019a*). Anatomically, the IPS is a major landmark on the lateral surface of the macaque parietal lobe. It separates the superior (SPL) from the inferior (IPL) lobule. Along the rostral-caudal direction, the IPS is strategically situated between the sensorimotor cortex around the central sulcus and the visual cortex in the occipital lobe.

Such a functional diversity raises the question of a corresponding anatomical organization. Most anatomical studies only relied on objective observation of cortical cytoarchitecture, leading to a diversity of parcellation schemes. Early cortical parcellations did not differentiate the IPS from the adjacent superior and inferior parietal lobule (*Brodmann, 1909*; *Vogt and Vogt, 1919*; *Von Bonin and Bailey, 1947*). Until *Seltzer and Pandya, 1986*; *Pandya and Seltzer, 1982*; *Seltzer and Pandya, 1980* distinguished IPS from SPL/IPL and separated it into three distinct architectonic zones. Here, areas PEa and POa occupied the medial and lateral bank respectively, and area Ipd was found

in the depth of the sulcus (*Figure 1A*). These large zones were further subdivided into up to nine smaller areas (*Figure 1B–D*; *Preuss and Goldman-Rakic, 1991*; *Lewis and Van Essen, 2000a*; *Bakola et al., 2017*). Although it is difficult to reach a complete consensus on the exact number and precise arrangement of the IPS areas, a rostro-caudally elongated shape of the areas along the lateral and medial banks as well as in the fundus of the IPS is a common feature in these maps.

Simultaneously, several functionally different regions within macaque IPS have been delineated based on their neuronal response properties or functional connectivities (*Bakola et al., 2017*; *Colby et al., 1988*; *Blatt et al., 1990*; *Matelli et al., 1998*; *Lewis and Van Essen, 2000b*; *Bakola et al., 2010*). In macaque monkeys, the lateral wall of IPS seems to be extensively interconnected with visual areas (*Colby et al., 1988*; *Maunsell and van Essen, 1983*; *Boussaoud et al., 1990*), whereas the medial wall has strong connections with somatosensory and somatomotor areas (*Sakata et al., 1997*; *Bakola et al., 2017*). Additionally, the cortical connections between the IPS and other cortical areas change from rostral to caudal, as demonstrated in several studies (*Gamberini et al., 2015*; *Felleman and Van Essen, 1991*). Specifically, areas located in the caudal IPS are the target of projections directly from primary visual and early extrastriate visual areas (*Gamberini et al., 2015*; *Gattass et al., 1988*), whereas areas belonging to the rostral IPS send projections to motor and premotor areas in the frontal lobe (*Lewis and Van Essen, 2000b*; *Romero and Janssen, 2016*). Thus, the structural, functional, and connectivity data favor the concept of a highly segregated mapping for macaque IPS.

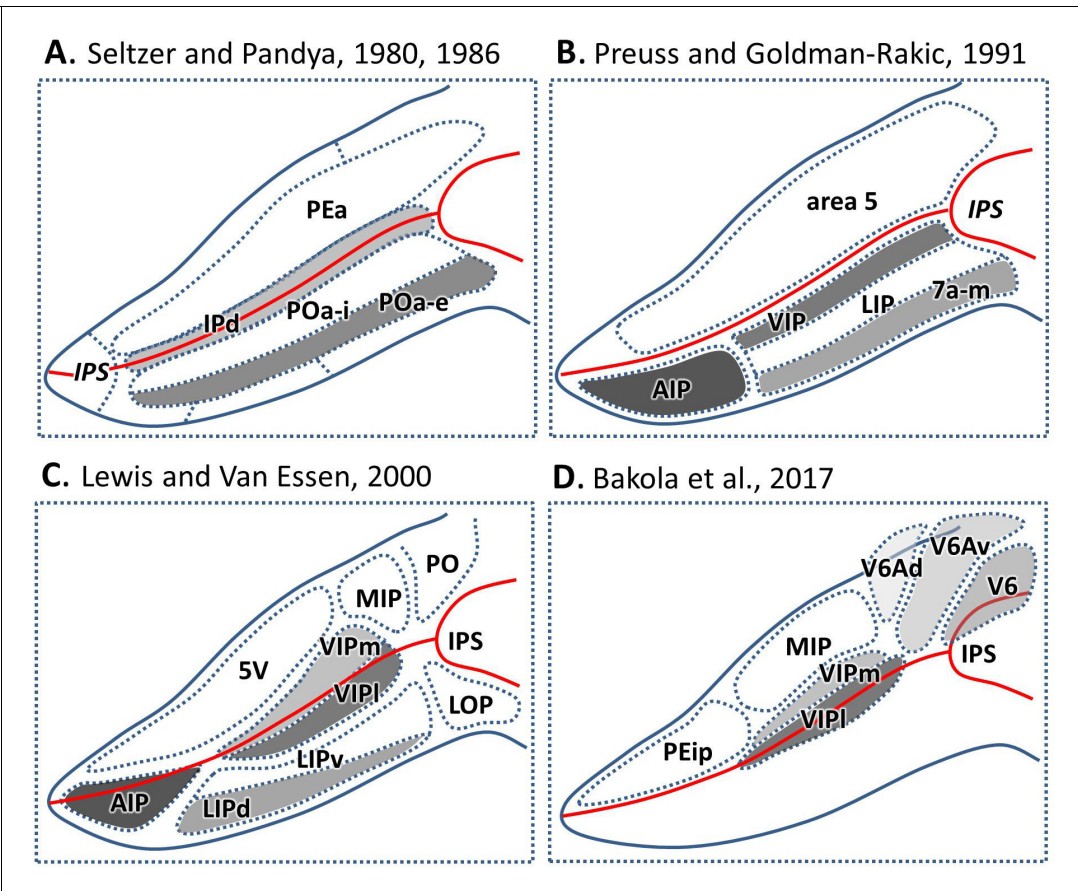

**Figure 1.** Parcellation of IPS areas in previous studies. The IPS is unfolded. Blue lines represent the outline of the unfolded IPS, red line represents the fundus of the IPS. (A) Map after *Seltzer and Pandya, 1986*, *Seltzer and Pandya, 1980*. (B) Map after *Preuss and Goldman-Rakic, 1991*. (C) Map after *Lewis and Van Essen, 2000a*. (D) Map after *Bakola et al., 2017*. Abbreviations: *IPS* intraparietal sulcus, *IPd* intraparietal area (deep), *PEa* parietal area PEa, *POa-i* internal part of area POa, *POa-e* external part of area POa, *AIP* anterior intraparietal area, *VIP* ventral intraparietal area, *VIPm* ventral intraparietal area (medial part), *VIPl* ventral intraparietal area (lateral part), *LIP* lateral intraparietal area, *LIPd* lateral intraparietal area (dorsal), *LIPv* lateral intraparietal area (ventral), *7a-m*, medial part of area 7a, *5V* ventral part of area 5, *MIP* medial intraparietal area, *PO* parietal-occipital area, *LOP* lateral occipital parietal, *PEip* intraparietal part of PE, *V6* visual area 6, *V6Av* visual area 6A (ventral part), *V6Ad* visual area 6A (dorsal part).

Receptors for classical neurotransmitters are heterogeneously distributed throughout cortical regions and layers. Thus, the visualization of multiple receptors in serial brain sections provides a quantitative and multimodal approach to detect borders between different cortical areas. Furthermore, within a given area, different receptor types differ in their absolute concentrations. Since receptors constitute key molecules in signal transmission, the simultaneous analysis of multiple receptors provides crucial information concerning the molecular basis of regional organization into functional networks (*Palomero-Gallagher and Zilles, 2018*; *Zilles and Amunts, 2009*; *Zilles et al., 2002a*). During the last two decades, the organization of several cortical regions in the human and monkey brain have been explored with this approach enabling a correction and refinement of existing parcellation schemes (*Zilles et al., 2004*; *Vogt et al., 2013*; *Palomero-Gallagher et al., 2019a*; *Impieri et al., 2019*; *Caspers et al., 2013*; *Palomero-Gallagher et al., 2008*; *Palomero-Gallagher and Zilles, 2019b*). Such multi-receptor studies also revealed characteristic differences in multiple receptor expression of different functional systems and hierarchical levels within a functional system (*Zilles et al., 2015*). It could be shown that cortical areas with similar multi-receptor expression ('receptor fingerprint') are parts of the same functional neural network (*Zilles and Amunts, 2009*; *Zilles et al., 2002a*; *Zilles and Palomero-Gallagher, 2017*).

Therefore, the goal of the present study is to explore the regional organization of the IPS and its junction with the parieto-occipital sulcus (POS) based on the regional and laminar distribution of the densities of multiple receptors combined with cyto- and myeloarchitecture. Here, we provide a refined map of the IPS based on this multimodal approach. Furthermore, analysis of the similarities and dissimilarities of the receptor fingerprints provides insight into the areal segregation and functional networks in the IPS.

## Results

Seventeen distinct subdivisions were cyto-/myelo- and receptor architectonically mapped within the macaque IPS and at its junction with the POS. The present multimodal study not only confirms the existence of some previously described areas, but also enables the definition of novel subdivisions within area MIP (i.e. dorsal and ventral parts of MIP), as well as within area PEip (i.e. external and internal parts of PEip). Each area has its own characteristic receptor expression pattern, and the similarities and dissimilarities of their receptor fingerprints enabled the segregation of IPS areas into to three functionally relevant groups.

### Topography of the cortical areas in or near the intraparietal sulcus

Eight cortical areas (V3d, V3A, PIP, LOP, V4d, V6, V6Ad, V6Av) were found at the junction of IPS with the POS (*Figures 2* and *3*). Area V3d lies on the annectant gyrus (*Figure 2*), which is laterally and medially bordered by the fundus of POS. PIP lies in the POS sulcal fundus rostro-medial to the annectant gyrus between V3d and the rostral parts of V6/V6Av. Areas V6, V6Av, and V6Ad are found on the rostral bank of the POS. V3A lies in the sulcal fundus lateral to the annectant gyrus at the same rostro-caudal level as V3d. V4d is situated on the posterior wall of prelunate gyrus. On the lateral bank of IPS, LOP is located between the rostral parts of V3A/V4d and LIPv.

In addition, we identified nine architectonically distinct areas completely buried in the IPS (*Figures 2* and *3*). Dorsal (MIPd) and ventral (MIPv) MIP cover the posterior part of the medial wall of the IPS. Both areas adjoin the superior parietal lobe dorsally, and abut V6Ad caudally. Rostral to MIPd and MIPv, PEip occupies the anterior two thirds of the medial bank of IPS. PEip can be subdivided into two regions based on differences in cyto- and receptor architecture, that is, external (PEipe) and internal PEip (PEipi). On the lateral bank of the IPS, the posterior two thirds, classically considered as LIP, contain two architectonically distinct areas: dorsal LIP (LIPd) and ventral LIP (LIPv). The fundus of IPS is occupied by the ventral intraparietal area VIP, which can be divided into a medial (VIPm) and a lateral part (VIPl). The most anterior part of IPS comprises areas AIP and the rostral portion of both VIP subdivisions. AIP is mainly located on the lateral bank of the anterior IPS, but when approaching the tip of IPS, it extends to the fundus of IPS and replaces VIP.

Based on these findings, we created a 2D flat map with detailed information concerning the location and extent of each cyto- and receptor architectonically identified IPS area. The 2D framework was created by unfolding the intraparietal sulcus, marking the position of its fundus and of the annectant gyrus. The parcellation of each brain was mapped in such a way, that the position of an

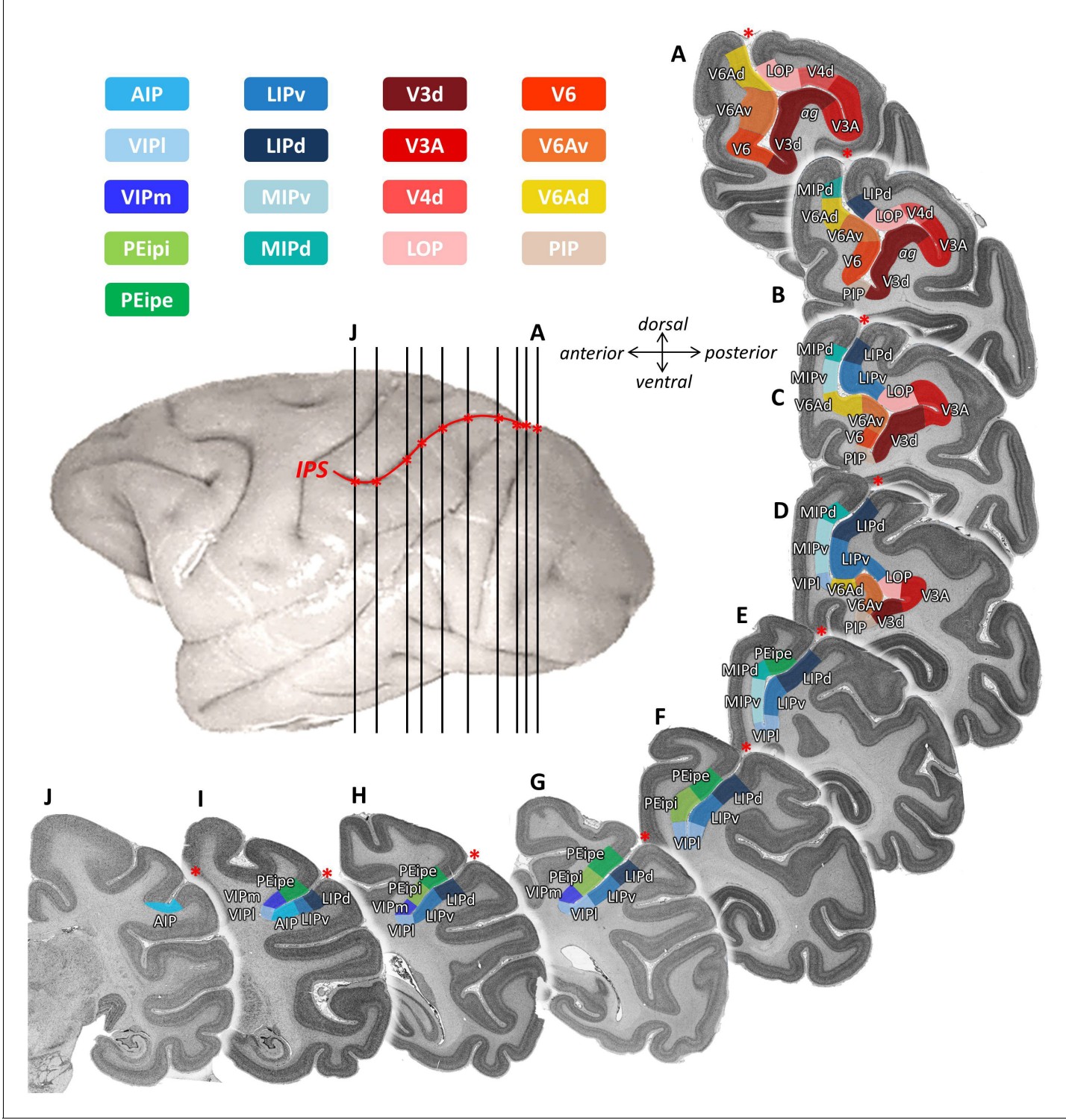

**Figure 2.** Topography of IPS areas, and at the junction with POS. Architectonic divisions are shown in a series of coronal cell body-stained sections from the left hemisphere of *Macaca mulatta* (DP1). The position of each section is highlighted on the lateral view of the left hemisphere. Abbreviations: *IPS* intraparietal sulcus, *POS* parietal-occipital sulcus, *AIP* anterior intraparietal area, *VIPm* ventral intraparietal area (medial part), *VIPl* ventral intraparietal area (lateral part), *PEipe* intraparietal part of PE (external part), *PEipi* intraparietal part of PE (internal part), *LIPd* lateral intraparietal area (dorsal), *LIPv* lateral intraparietal area (ventral), *MIPd* medial intraparietal area (dorsal), *MIPv* medial intraparietal area (ventral), *V3d* dorsal part of visual area 3, *V3A* visual area 3A, *V4d* dorsal part of visual area 3, *LOP* lateral occipital parietal, *V6* visual area 6, *V6Av* visual area 6A (ventral part), *V6Ad* visual area 6A (dorsal part), *PIP* posterior intraparietal area.

areal border could be traced relative to the macroscopical landmarks, that is fundi and sulci. Thus, the information concerning the location of architectonic areas could be conveniently displayed in a simple visualization format. As an example, *Figure 3* shows the 2D cytoarchitectonic parcellation in both hemispheres of the macaque brain DP1.

## Cyto- and myeloarchitecture of IPS areas

Nomenclature and criteria for cytoarchitectonic mapping were based on previous studies (*Seltzer and Pandya, 1986*; *Pandya and Seltzer, 1982*; *Seltzer and Pandya, 1980*; *Preuss and Goldman-Rakic, 1991*; *Lewis and Van Essen, 2000a*; *Bakola et al., 2017*; *Luppino et al., 2005*). The cytoarchitectonic features of most areas found in the present study are comparable to those of previous studies. *Figure 4* shows the laminar pattern of these areas in the present study. Since we concur with the cytoarchitectonic descriptions of these areas in the literature, we will add here a detailed description only of the cytoarchitecture of the newly defined areas, that is MIPd, MIPv, PEipe and PEipi.

Areas MIPd and MIPv almost completely occupy the posterior of medial wall of the IPS. The border between them was confirmed by the statistical analysis of both the cyto- (*Figure 5*) and receptor architecture (*Figure 6*) of the portion of the cortical ribbon classically attributed to area MIP. Area MIPd covers the upper one-third of medial wall and is characterized by a clear delineation of all layers. Area MIPv is found on the lower part of the IPS and can be recognized by its cell-sparse layer

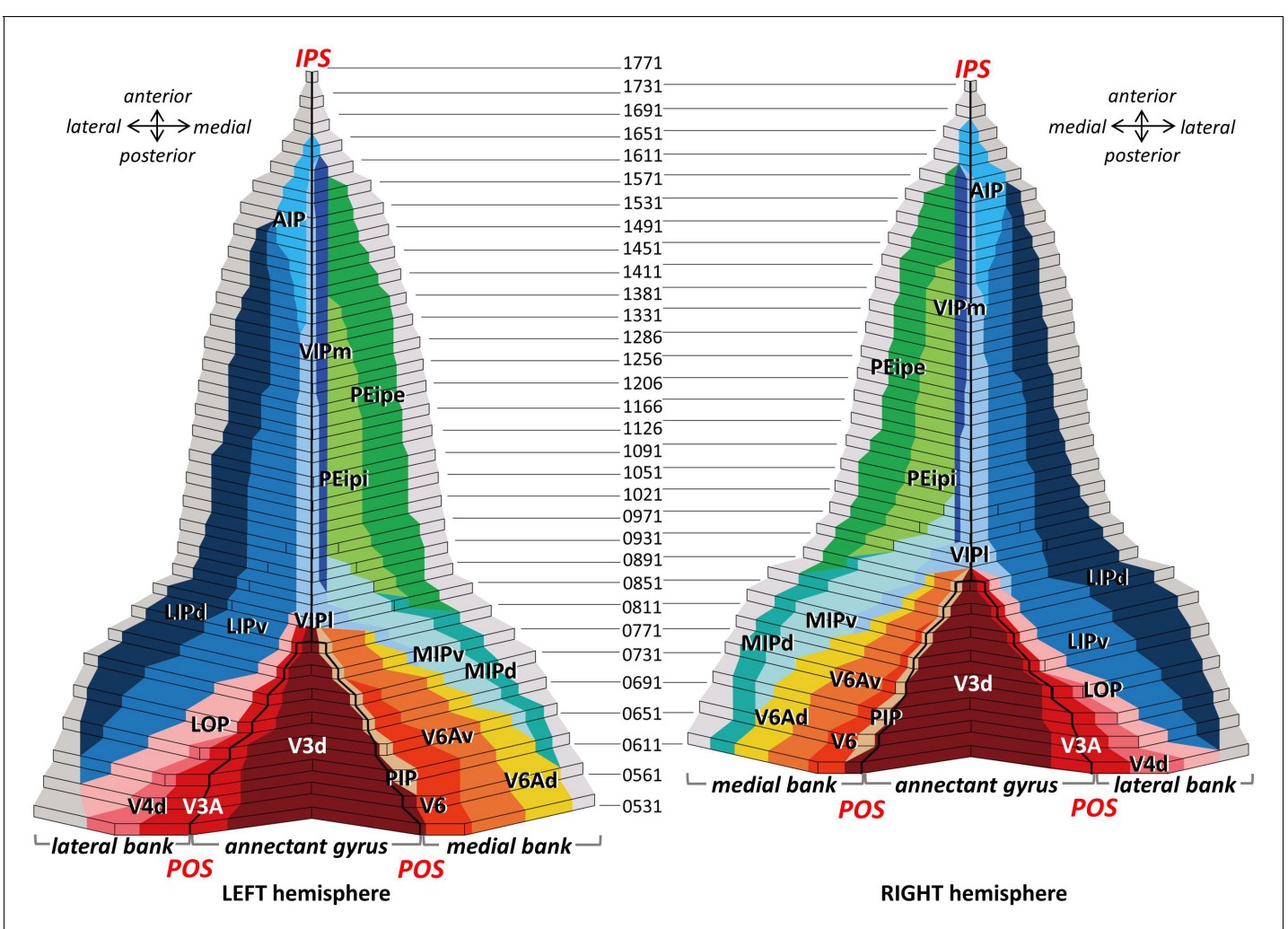

**Figure 3.** 2D cytoarchitectonic flat map of IPS areas in both hemispheres of Macaca mulatta. Bold lines represent fundi of the intraparietal (IPS) and parieto-occipital (POS) sulci. Arabic numerals mark section numbers. For abbreviations see *Figure 2*.

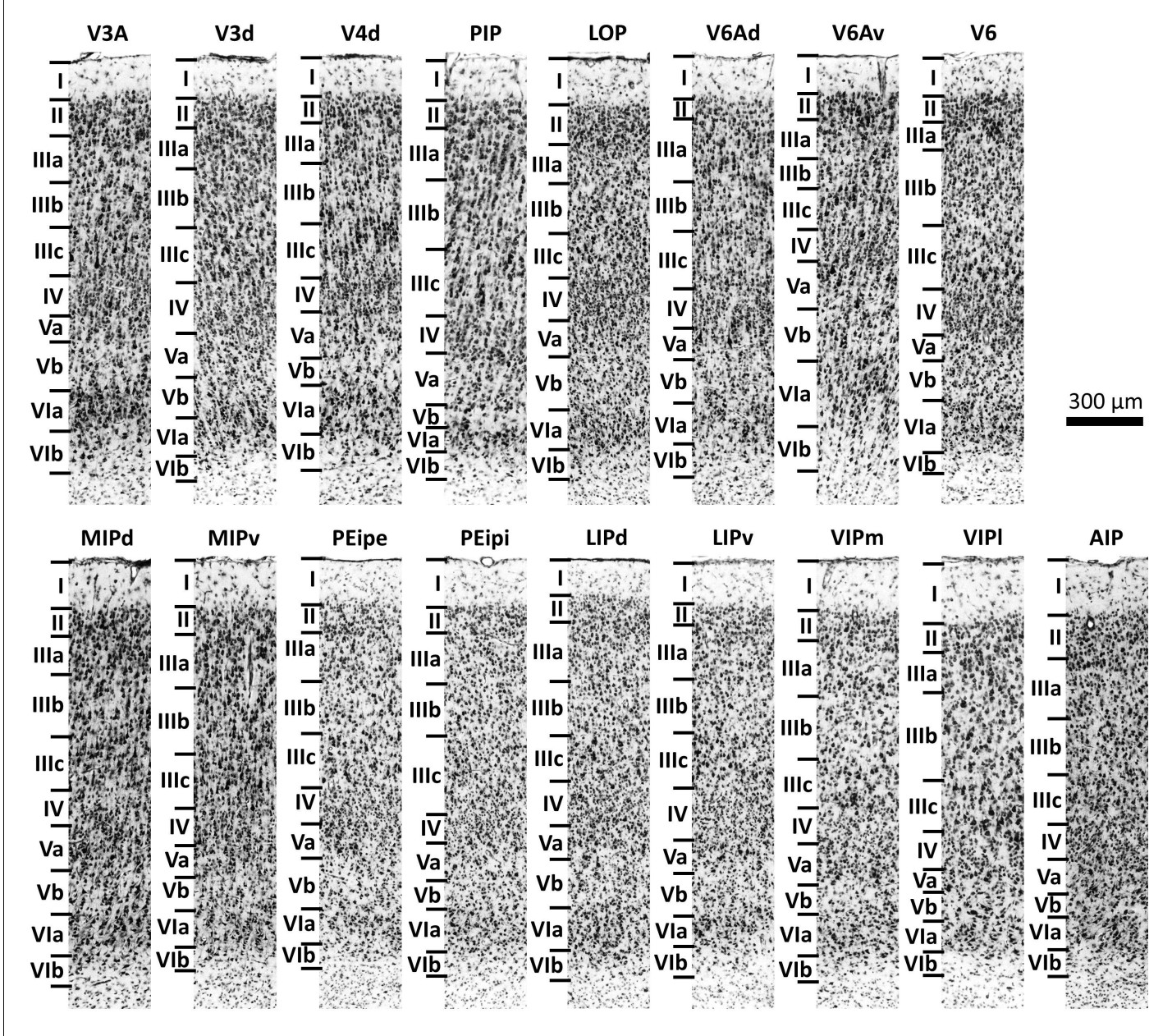

**Figure 4.** Cytoarchitecture of the 17 IPS areas. Scale bar 300 μm. Roman numerals indicate cytoarchitectonic layers. For abbreviations, see *Figure 2*.

IIIc with some scattered very large pyramidal cells and the broad layer Vb with a low cell packing density. Compared to area MIPv, MIPd has narrower layers III and Vb with a higher cell packing density, whereas layer VIa is much thinner in MIPv than in MIPd (*Figure 5D*).

PEip occupies the anterior two-thirds of the medial bank of the IPS. It is a rostro-caudally oriented strip of cortex, which has previously been described as architectonically homogenous. However, in the coronal planes of the present study, external (PEipe) and internal (PEipi) subdivisions could be identified by visual inspection and confirmed by means of the observer-independent analysis (*Figures 7* and *8*). As shown in *Figure 7D*, layer II of both areas is difficult to separate from layer III. In both areas, layers IIIb and IIIc can be distinguished by the lower cell density in IIIb, and the presence of large pyramidal cells in IIIc. Layer Vb appears as a pale stripe between Va and the densely populated layer VIa. However, compared to PEipe, layers IIIa and IIIb of PEipi are very similar in their cell packing density, and cannot be clearly separated. The layer IIIc pyramidal cells in PEipi are smaller

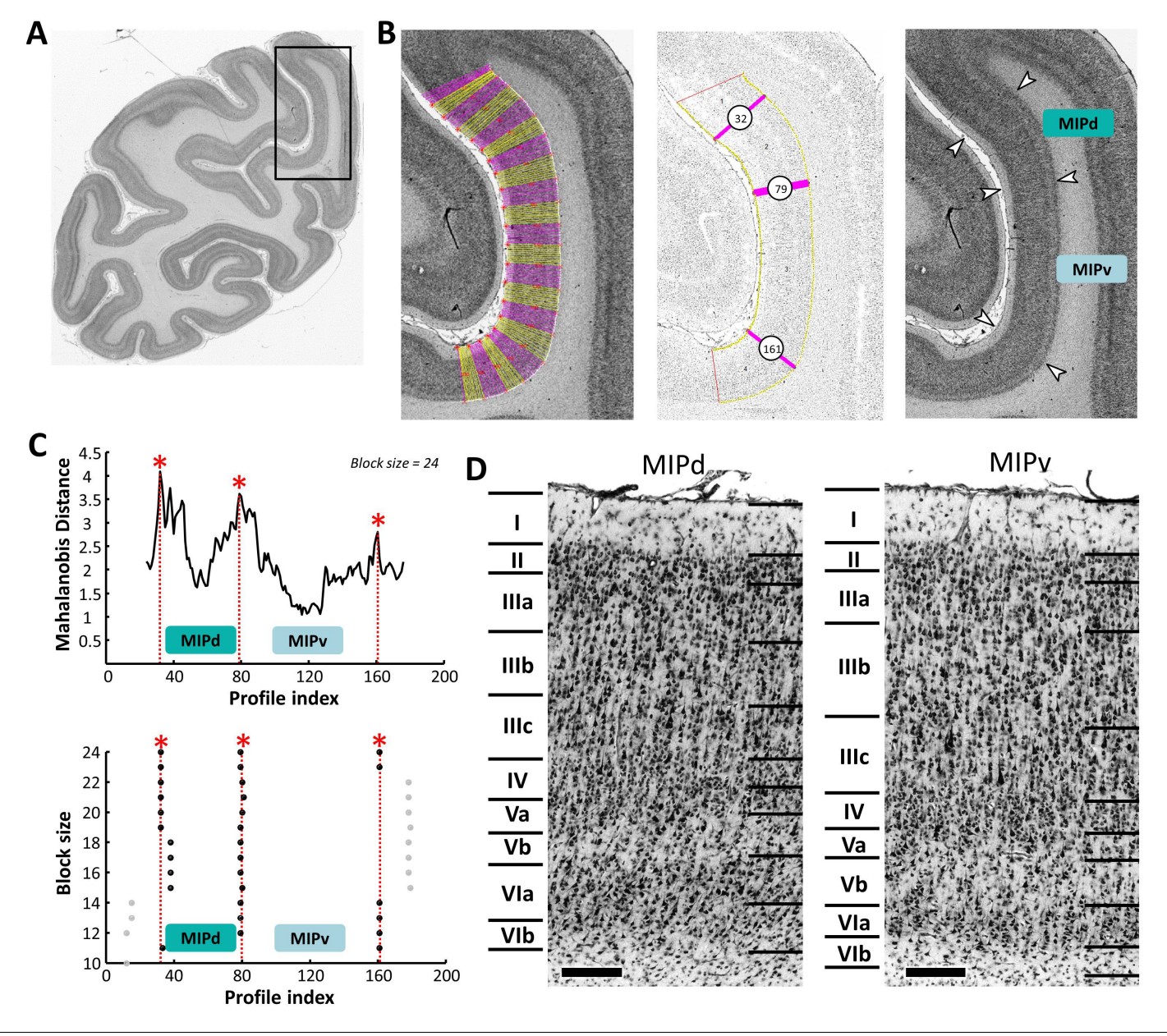

**Figure 5.** Example of the identification of cytoarchitectonic borders for MIPd and MIPv. (**A**) The black box on the overview of a coronal section processed for cell bodies indicates the location of the photomicrograph shown on the right. (**B**) The cortical ribbon was covered by traverses (from which GLI profiles were extracted) running perpendicular to the cortical layers; color changes between yellow and pink after every 10th traverse (left). Automatic labeling of the positions of the statistically defined borders (at position 32, 79 and 161) (middle). The position of borders of areas MIPd and MIPv as defined by visual inspection are indicated by white arrows in the corresponding image of the cell body stained section (right). (**C**) Exemplary Mahalanobis distance function depicting the Mahalanobis distances between neighboring blocks of 24 profiles; significant maxima occurred at profile positions 32, 79 and 161, which identify the borders of areas MIPd and MIPv (top). The significant maxima of varying block sizes (ranging from 10 to 24), indicate a consistently occurring border between both areas at profile location 79 (bottom). (**D**) High-resolution photomicrographs of cytoarchitectonic subdivisions MIPd and MIPv. Scale bar 200 μm. Roman numerals indicate cytoarchitectonic layers. Abbreviations: *MIPd* medial intraparietal area (dorsal), *MIPv* medial intraparietal area (ventral).

than those in PEipe; though still present in considerable numbers, they do not stand out as prominently as in PEipe. Layer IV of PEipi is somewhat broader than in PEipe. Layer Va of PEipi is broader and more easily detectable than in PEipe, where the thin layer Va merges with layer IV. In general, PEipe shows a clearer lamination than does PEipi (*Figure 7D*).

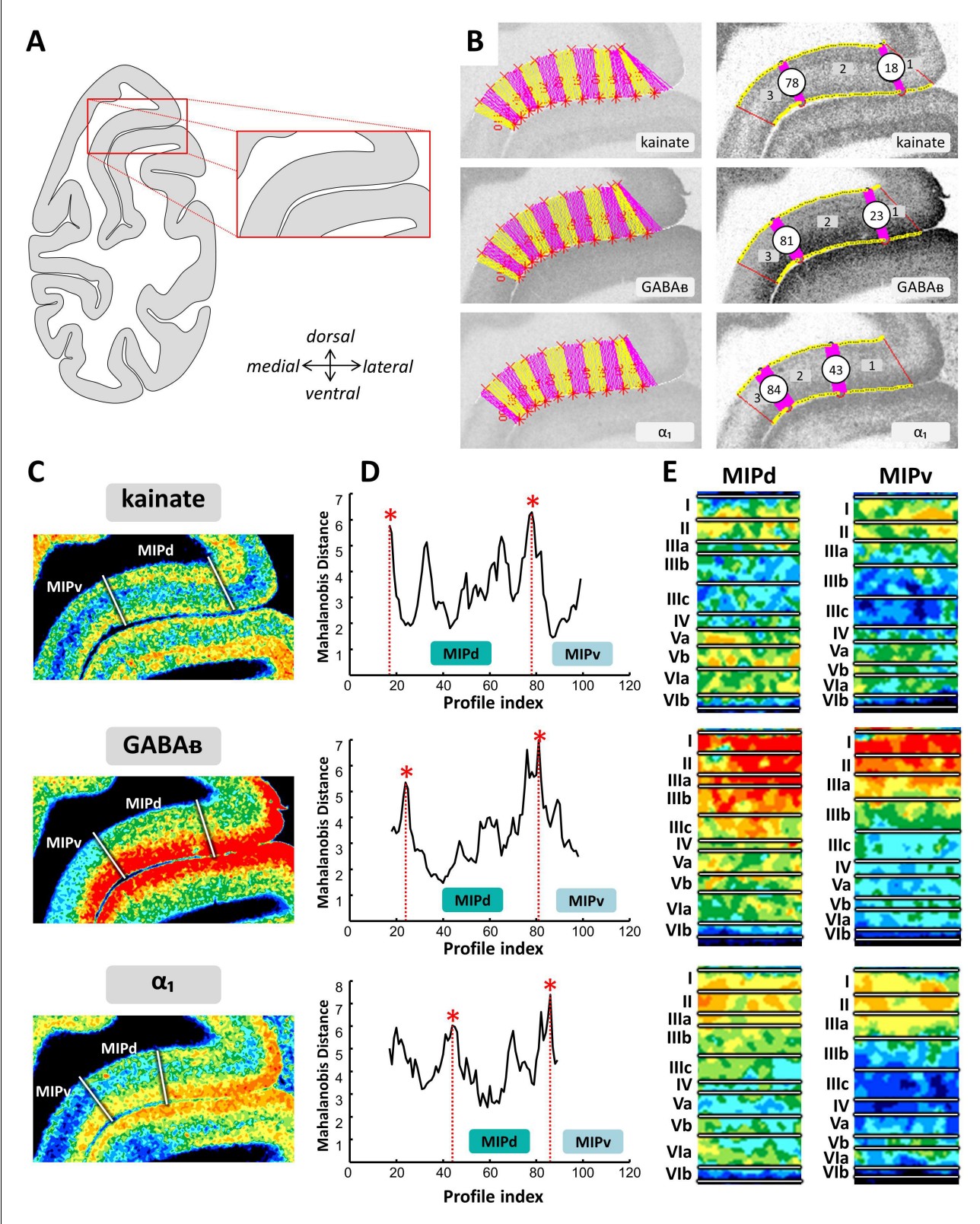

**Figure 6.** Example of the identification of area MIPd based on receptor distribution patterns. (**A**) Schematic representation of an autoradiograph in which the red box indicates the location of the exemplary autoradiographs (kainate, GABA_B and $\alpha_1$ receptors) shown in (**B**) and (**C**). (**B**) The cortical ribbon was covered by traverses (from which profiles were extracted) running perpendicular to the cortical layers; color changes between yellow and pink after every 10th traverse (left). Automatic labeling of the positions of the statistically defined borders (right). (**C**) The position of borders of area

*Figure 6 continued on next page*

*Figure 6 continued*

MIPd as defined by visual inspection are indicated by white lines in corresponding pseudocolor coded autoardiographs. (D) Exemplary Mahalanobis distance function depicting the Mahalanobis distances between neighboring blocks of 20 profiles; significant maxima occurred at profile positions which coincide with borders for receptor architectonically defined area MIPd. (E) Laminar distribution patterns of corresponding receptors in areas MIPd and MIPv. Roman numerals indicate cytoarchitectonic layers. Abbreviations: *MIPd* medial intraparietal area (dorsal), *MIPv* medial intraparietal area (ventral).

The online version of this article includes the following figure supplement(s) for figure 6:

**Figure supplement 1.** Cyto-, myelo- and receptor architecture of macaque MIPd.

**Figure supplement 2.** Cyto-, myelo- and receptor architecture of macaque MIPv.

In myeloarchitecture (*Figure 9*), both MIPd and MIPv have a thin inner band of Baillarger and a densely packed broad outer band of Baillarger. Area MIPv could be distinguished from MIPd only by its relatively longer and more prominent vertical fiber bundles, which reach into the superficial layers. The myeloarchitecture of PEipi (*Figure 9*) shows equally prominent and dense inner and outer Baillarger stripes, whereas PEipi shows a relatively lighter myelination and has a slightly more prominent outer than inner Baillarger stripe.

## Receptor mapping

We used the Macaca fascicularis monkeys #11530 (left hemisphere), #11539 (left and right hemispheres) and #11543 (left hemisphere) for the receptor architectonic analysis. *Figure 10* shows a series of coronal sections from posterior to anterior labeled for the demonstration of representative receptors. Differences in the densities and laminar distribution patterns of the analyzed receptors-enabled delineation of the different cortical areas in the IPS. The receptor-based parcellation approach led to the identification of the same 17 areas as determined by cytoarchitecture. As an example for the observer-independent definition of areal borders of macaque IPS, the receptor architectonically defined borders for different receptor types between newly defined areas MIPd, PEipe and PEipi shown in *Figures 6* and *8*.

## Areas located at the junction of the IPS and POS

Receptor distribution patterns of the three areas which are located in the proximity of the POS, that is, areas V6, V6Av, and V6Ad, are shown in *Figure 10* and *Figure 10—figure supplement 1*. Area V6 contains significantly lower mean (averaged over all cortical layers) $\alpha_1$ receptor densities than does V6Av, as well as significantly lower AMPA, $\alpha_1$, 5-HT$_{1A}$, and D$_1$ receptor densities than does V6Ad. Furthermore, V6Av contains significantly lower 5-HT$_{1A}$ receptor densities than does V6Ad. Notably, V6Ad presented the highest densities of most receptors, V6Av intermediate values, and V6 the lowest ones, whereas the opposite trend was found for M$_2$ receptor densities, which were highest in V6 and lowest in V6Ad. Furthermore, the clear bilaminar distribution of M$_2$ receptors in V6Ad and V6Av enabled delineation of both V6A areas from adjacent areas V6 and MIPd.

Area V3d could be distinguished from the surrounding areas mainly by its distinctive receptor distribution patterns in the infragranular layers (*Figure 10* and *Figure 10—figure supplement 1*). Higher AMPA, kainate, M$_2$, and $\alpha_1$ receptor densities, and lower NMDA and GABAergic receptor densities were found in the infragranular layers of V3d than in those of surrounding areas. Receptors for GABA provided evidence for the identification of area PIP by their higher densities in the deep layers of area PIP than in those of adjacent areas.

V4d showed slightly higher concentrations of kainate, GABA$_A$/BZ, M$_2$, $\alpha_1$ and 5-HT$_2$ receptors than neighboring V3A. In addition, V4d was distinguishable from LOP mainly due to its higher densities in GABA$_A$, GABA$_B$, GABA$_A$/BZ and 5-HT$_2$ receptors, as well as the lower densities in A$_1$ receptors, especially in the deeper layers (*Figure 10* and *Figure 10—figure supplement 1*). Areas LOP and LIPv could be separated by the lower concentrations of AMPA, GABA$_A$, 5-HT$_{1A}$, and higher concentrations of $\alpha_2$ receptors in the former than in the latter area (*Figure 10* and *Figure 10—figure supplement 1*).

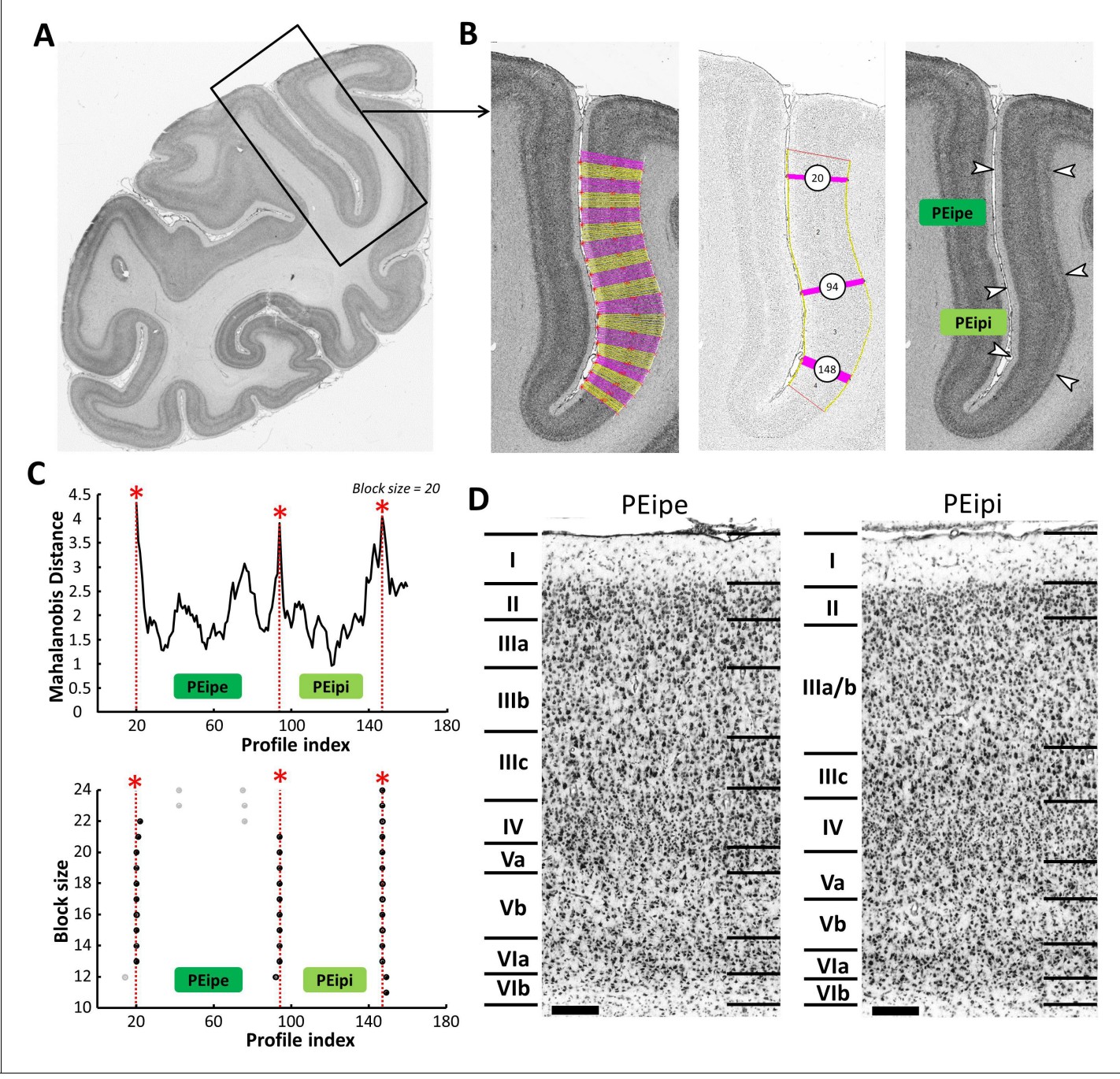

**Figure 7.** Example of the identification of cytoarchitectonic borders for PEipe and PEipi. (**A**) The black box on the overview of a coronal section processed for cell bodies indicates the location of the photomicrograph shown on the right. (**B**) The cortical ribbon was covered by traverses (from which GLI profiles were extracted) running perpendicular to the cortical layers; color changes between yellow and pink after every 10th traverse (left). Automatic labeling of the positions of the statistically defined borders (at position 20, 94 and 148) (middle). The position of borders of areas PEipe and PEipi as defined by visual inspection are indicated by white arrows in the corresponding image of the cell body stained section (right). (**C**) Exemplary Mahalanobis distance function depicting the Mahalanobis distances between neighboring blocks of 20 profiles; significant maxima occurred at profile positions 20, 94 and 148, which identify the borders of areas PEipe and PEipi (top). The significant maxima of varying block sizes (ranging from 10 to 24), indicate a consistently occurring border between both areas at profile location 94 (bottom). (**D**) High-resolution photomicrographs of cytoarchitectonic subdivisions PEipe and PEipi. Scale bar 200 μm. Roman numerals indicate cytoarchitectonic layers. Abbreviations: *PEipe* intraparietal part of PE (external part), *PEipi* intraparietal part of PE (internal part).

eLife Research article    Neuroscience

**Figure 8.** Example of the identification of areas PEipi and PEipe based on receptor distribution patterns. (**A**) Schematic representation of an autoradiograph in which the red box indicates the location of the exemplary autoradiographs (M₂ M₃ and α₁ receptors) shown in (**B**) and (**C**). (**B**) The cortical ribbon was covered by traverses (from which profiles were extracted) running perpendicular to the cortical layers; color changes between yellow and pink after every 10th traverse (left). Automatic labeling of the positions of the statistically defined borders (right). (**C**) The position of borders of

*Figure 8 continued on next page*

*Figure 8 continued*

areas PEipi and PEipe as defined by visual inspection are indicated by white lines in corresponding pseudocolor coded autoardiographs. (D) Exemplary Mahalanobis distance function depicting the Mahalanobis distances between neighboring blocks of 20 profiles; significant maxima occurred at profile positions which coincide with borders for receptor architectonically defined areas PEipi and PEipe. (E) Laminar distribution patterns of corresponding receptors in areas PEipi and PEipe. Roman numerals indicate cytoarchitectonic layers. Abbreviations: *PEipe* intraparietal part of PE (external part), *PEipi* intraparietal part of PE (internal part).

The online version of this article includes the following figure supplement(s) for figure 8:

**Figure supplement 1.** Cyto-, myelo- and receptor architecture of macaque PEipe.
**Figure supplement 2.** Cyto-, myelo- and receptor architecture of macaque PEipi.

## Areas located in the IPS

The caudal part of the lateral wall of IPS is occupied by LIPv and LIPd (*Figure 10* and *Figure 10—figure supplements 2–3*). These two areas differ significantly in their mean NMDA, $\alpha_1$ and 5-HT$_{1A}$ receptor densities, which are higher in LIPd compared to LIPv.

Three areas are located in the depth of the sulcus: along most of the fundus of the IPS, we defined medial and lateral portions of VIP; AIP is located on the most anterior part of the fundus and extends onto the lateral bank of the IPS. Although the receptor architectonic differences between VIPl and VIPm are subtle, VIPl could be distinguished from VIPm by its slightly thicker cortical ribbon, as well as the more apparent laminar distribution pattern of AMPA, kainate, M$_2$, M$_3$ and $\alpha_1$ receptors (*Figure 10* and *Figure 10—figure supplement 3*). Area AIP can be distinguished from its caudally neighboring areas VIPm and VIPl by its significantly higher $\alpha_1$ and 5HT$_{1A}$ receptor densities. Although the receptor architecture of area AIP resembled that of the neighboring somatosensory area 2 and inferior parietal area PF, the densities of kainate, 5-HT$_{1A}$, 5-HT$_2$ and A$_1$ receptors were lower in the infragranular layers of AIP than in those of adjacent areas. GABA$_A$ and GABA$_B$ receptor densities of AIP were lower than in area 2, but higher than in area PF. Finally, the D$_1$ receptor reached higher densities in the infragranular layers of AIP than in those of areas 2 and PF (*Figure 10* and *Figure 10—figure supplement 4*).

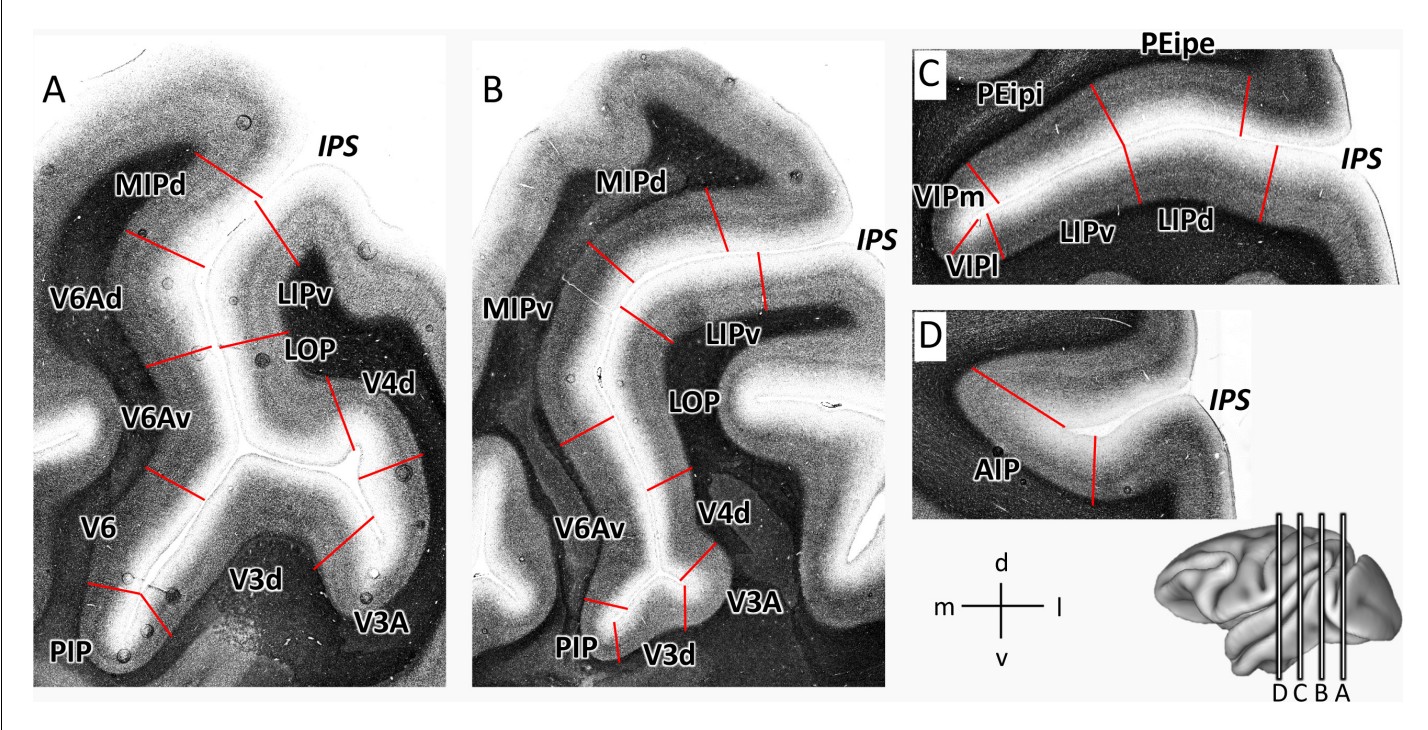

**Figure 9.** Myeloarchitecture of areas of the IPS. Coronal sections through four rostro-caudal levels of a macaque hemisphere showing the myeloarchitecture of the IPS. For abbreviations, see *Figure 2*.

On the medial wall of the posterior portion of the IPS, area MIPv had lower densities of all receptors (except $\alpha_2$) compared to MIPd (*Figure 10* and *Figure 10—figure supplement 2*). *Figure 6* and *Figure 6—figure supplements 1–2* show the laminar distribution of the fifteen receptors analyzed in MIPd and MIPv. Most receptors (except kainate, $M_2$, 5-$HT_2$, $\alpha_1$ and $A_1$ receptors) had a unimodal distribution of their laminar concentrations, with higher densities in the supragranular than in the infragranular layers. Only the 5-$HT_2$ and $A_1$ receptors reached relatively higher densities in the middle and deeper layers of MIPd. A bimodal distribution was found for kainate, $M_2$ and $\alpha_1$ receptors (*Figure 6* and *Figure 6—figure supplements 1–2*). Despite the comparable laminar distribution patterns in both areas, the differences in receptor densities between cortical layers were more apparent in MIPv than in MIPd. Furthermore, MIPd could be distinguished from area MIPv most clearly by differences in $\alpha_1$ receptor densities, which were significantly higher in the former area (*Figure 6*, *Figure 10* and *Figure 10—figure supplement 2*).

The rostral portion of the medial wall of the IPS contained two subdivisions of PEip, which differed in their receptor patterns (*Figure 8*, *Figure 10* and *Figure 10—figure supplement 3*). Most of the receptors (NMDA, $GABA_A$, $GABA_B$, $GABA_A$/BZ, $M_1$, $M_3$, $\alpha_2$, 5-$HT_{1A}$) show a unimodal laminar distribution pattern with decreasing densities from layers I or II to the border between VIa and VIb. Exceptions are found for kainate, $M_2$, $\alpha_1$, $A_1$ and $D_1$ receptors, which are bimodally distributed in PEipe and PEipi (*Figure 8* and *Figure 8—figure supplements 1–2*). Areas PEipe and PEipi differ in their $M_2$ laminar distribution patterns (*Figure 8*, *Figure 10* and *Figure 8—figure supplements 1–2*). In PEipe, $M_2$ receptor densities are higher in layer V than in layer III, however, these layers present comparable $M_2$ receptor densities in PEipi (*Figure 8* and *Figure 8—figure supplements 1–2*).

## Multivariate analysis of mean receptor densities

*Figure 11* shows the receptor fingerprints of all areas (except V4d) studied here. The absolute receptor densities are averaged over all cortical layers (*Table 1*). Thus, the fingerprints represent the area-specific balances between the 15 receptors. Due to oblique planes of sectioning in two of the four brains processed for receptor autoradiography, it was only possible to measure receptor densities in area V4d of other two hemispheres. Thus, we did not create a fingerprint for area V4d. Highest absolute densities were reached by the NMDA, $GABA_A$, and $GABA_B$ receptors, as well as by the $GABA_A$/BZ-binding sites, and lowest absolute densities were measured for the $M_2$ and $D_1$ receptors (*Figure 11*).

Visual comparison of the fingerprints revealed a basic similarity between the areas V3d, V3A, PIP, LOP, V6, V6Av and V6Ad, whereas the areas AIP, MIPv, MIPd, PEipi, PEipe, LIPd, LIPv, VIPm and VIPl had fingerprints different from the previous areas mainly because of the higher density of the $GABA_B$ receptors. Additionally, receptor fingerprints seemed to be slightly larger in the areas near to the brain surface (V6Ad, MIPd, PEipe and LIPd) than their counterparts (V6, V6Av, MIPv, PEipi and LIPv) located near to the depth of sulcus.

In order to analyze quantitatively the degree of (dis)similarity of the receptor fingerprint between all areas studied here, a hierarchical cluster analysis was performed (*Figure 12A*). A k-means analysis showed that the fingerprints can be separated into three branches: Branch 1 of the cluster tree comprised the areas located on both walls of the IPS (i.e. MIPv, MIPd, LIPv, LIPd, PEipe, PEipi and AIP; *Figure 12B*); branch 2 contained the areas which are located on the IPS/POS junction, together with both subdivisions of VIP. Branch 2.1 contained areas VIPm, VIPl and V6Ad, and branch 2.2 comprised areas V6, V6Av, V3A, V3d, LOP and PIP (*Figure 12A*). The segregation of branches 1 and 2 was also confirmed by the 1st principal component of the PCA, and that of branches 2.1 and 2.2 by the 2nd principal component (*Figure 12C*).

## Discussion

Several maps of the macaque cortex in the IPS and at the confluence with POS have been published (*Seltzer and Pandya, 1986*; *Seltzer and Pandya, 1980*; *Preuss and Goldman-Rakic, 1991*; *Lewis and Van Essen, 2000a*; *Bakola et al., 2017*), but they differ in the number and precise extent of identified architectonical areas. Furthermore, it is sometimes not clear which areas are homologue; the different nomenclatural systems add further uncertainties because it is not always clear whether different names indicate different areas. Here, we will compare previous maps with present observations based on the similarity of the topographical relations. By combining information from

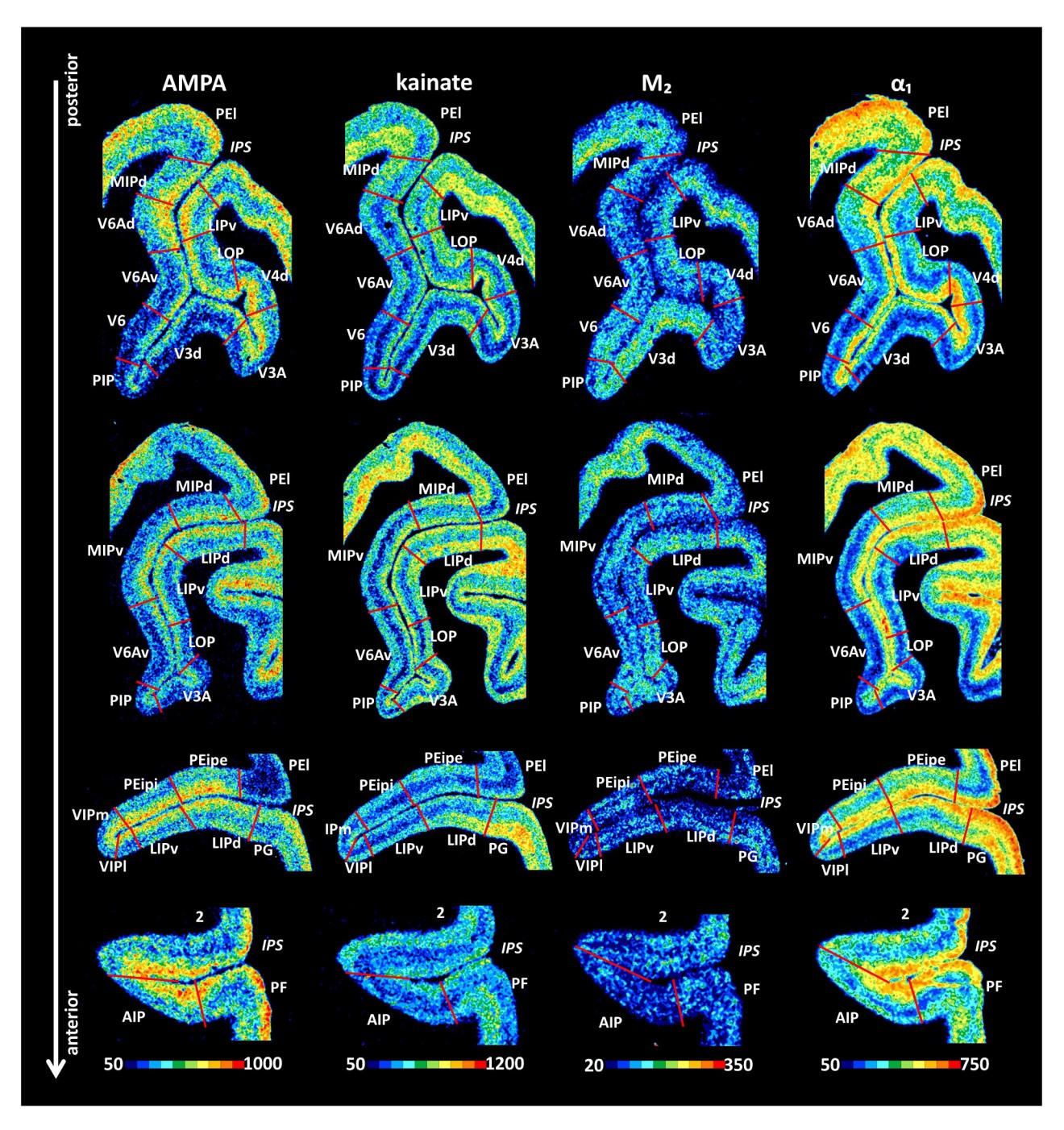

**Figure 10.** Example of receptor distribution patterns in the IPS. Coronal sections through four rostro-caudal levels of a macaque hemisphere showing exemplary receptor distribution patterns in the IPS. Receptor distribution patterns illustrated for all 15 receptors are shown in *Figure 10—figure supplements 1–4*. The borders between the IPS areas (red lines) are charted on the pseudocolor-coded autoradiographs. The color bar beneath each autoradiograph indicates receptor concentrations by the different colors, from black for low to red for high concentrations (fmol/mg protein). For abbreviations, see *Figure 2*.

The online version of this article includes the following figure supplement(s) for figure 10:

**Figure supplement 1.** Receptor distribution patterns in the junction of the IPS and POS.
**Figure supplement 2.** Receptor distribution patterns in the caudal IPS.
**Figure supplement 3.** Receptor distribution patterns in the middle portion of the IPS.
**Figure supplement 4.** Receptor distribution patterns in the rostral portion of the IPS.

cyto-, myeloarchitecture and multiple receptor architecture, the present study provides a complete multimodal architectonic map of the macaque IPS and at its junction with POS. Additionally, we present a 2D map with the explicit position of area borders and their relationship with macroanatomical landmarks, which allows comparison of the present parcellation with those from previous studies.

## Areas located at the junction of the IPS and POS

The junction of the IPS and POS is a convoluted region of the monkey brain that is involved in various functions. We were able to identify eight cyto-/myelo- and receptor architectonically distinct areas within this region of the macaque brain. Area V3d is located on the annectant gyrus, V3A and PIP are situated in the lateral and medial fundus of the POS, respectively. Area V6 covering the lower one-third of the rostro-medial wall of the POS, and the other two-thirds were occupied by the two subdivisions of V6A: V6Ad and V6Av. On the rostro-lateral bank of the POS, we confirmed the existence of LOP, which could be identified as a distinct area between V3A and LIPv. Although the main portion of area V4d is located on the posterior wall of the prelunate gyrus, which is not the topic of the present study, we also identified a small portion which encroached onto the rostro-lateral wall of POS.

Most parcellation schemes agree on the existence of areas V3A, V3d and PIP in the fundus of the IPS/POS junction (*Lewis and Van Essen, 2000a*; *Felleman et al., 1997*; *Nakamura et al., 2001*; *Ungerleider et al., 2008*; *Saleem and Logothetis, 2012*), and we were able to confirm these observations. However, a consensus has not yet been reached concerning the exact position of areas V3d and V3A in relation to the anectant gyrus. These discrepancies are probably due to the reported inter-individual variability in the width of area V3d (*Gattass et al., 1988*; *Gattass et al., 2005*; *Hadjidimitrakis et al., 2019b*) combined with the macroanatomical characteristics of this region of the macaque brain. Thus, depending on the relative width of area V3d, its border with V3A can be found either on the rostral wall of the anectant gyrus (*Lewis and Van Essen, 2000a*; *Figure 13A*), on the apex of the gyrus (*Saleem and Logothetis, 2012*; *Figure 13B*), or on its posterior wall (*Nakamura et al., 2001*; *Ungerleider et al., 2008*; *Figure 13C*). Although in parasagittal sections it is obvious that these three positions are only separated by 2–3 mm in the rostro-caudal direction (*Figure 13*), the variability results in apparent identification discrepancies when comparing studies based on coronally sectioned brains. A V3d/V3A border located on the rostral wall of the anectant gyrus would result in area V3d being found on the gyrus throughout most of its rostro-caudal extent, and V3A only encroaching onto its lateral surface at most rostral levels (*Lewis and Van Essen, 2000a*; *Figure 13A1–A3*). If the V3d/V3A border is located on the apex of the anectant gyrus, the most rostral sections will only contain V3A, the most caudal ones only V3d, and both areas will occupy approximately the same proportion of the gyrus at intermediate levels (*Saleem and Logothetis, 2012*; *Figure 13B1–B3*). Finally, if the V3d/V3A border is identified in the caudal wall of the antectant gyrus, then area V3A will be found on the gyrus throughout the first two-thirds of its rostro-caudal extent, and will be replaced by V3d at caudal levels (*Nakamura et al., 2001*; *Ungerleider et al., 2008*; *Figure 13C1–C3*). The position of the V3A/V3d border in present parcellation scheme is more similar to that charted by *Lewis and Van Essen, 2000a*, since for all examined hemispheres, the V3d/V3A border was found on the rostral wall of the anectant gyrus.

On the lateral bank, there is discrepancy related to the existence of area LOP, which is the cytoarchitectonic correlate of functionally identified caudal intraparietal area CIP (*Katsuyama et al., 2010*). *Lewis and Van Essen, 2000a*, first identified LOP based on its architectonic features, however, Schall, Morel (*Schall et al., 1995*) seem more inclined to include it as part of architectonically defined LIPv. On the other hand, differences between LOP/CIP and LIPv have been reported by connectional and physiological studies (*Lewis and Van Essen, 2000b*; *Tsutsui et al., 1999*). In accordance with the description of Lewis and Van Essen (*Lewis and Van Essen, 2000a*), we found LOP to have a broader and denser layer IV compared to LIPv; whereas LIPv shows larger and more prominent pyramidal cells in layer III and V. In addition, areas LOP and LIPv can be also delineated by the difference in distribution patterns of most receptors except for kainate, $GABA_A$, $M_1$ and $A_1$. Specifically, LOP shows considerably lower concentrations of AMPA, $GABA_B$, $\alpha_1$, $5\text{-}HT_{1A}$, $5\text{-}HT_2$ and $D_1$ receptors, although a slightly higher concentration can be noticed for NMDA, $M_3$, $\alpha_2$, and $5HT_{1A}$ receptors. In general, the present study provides evidence to confirm that LOP/CIP could be designated unequivocally as a distinct area.

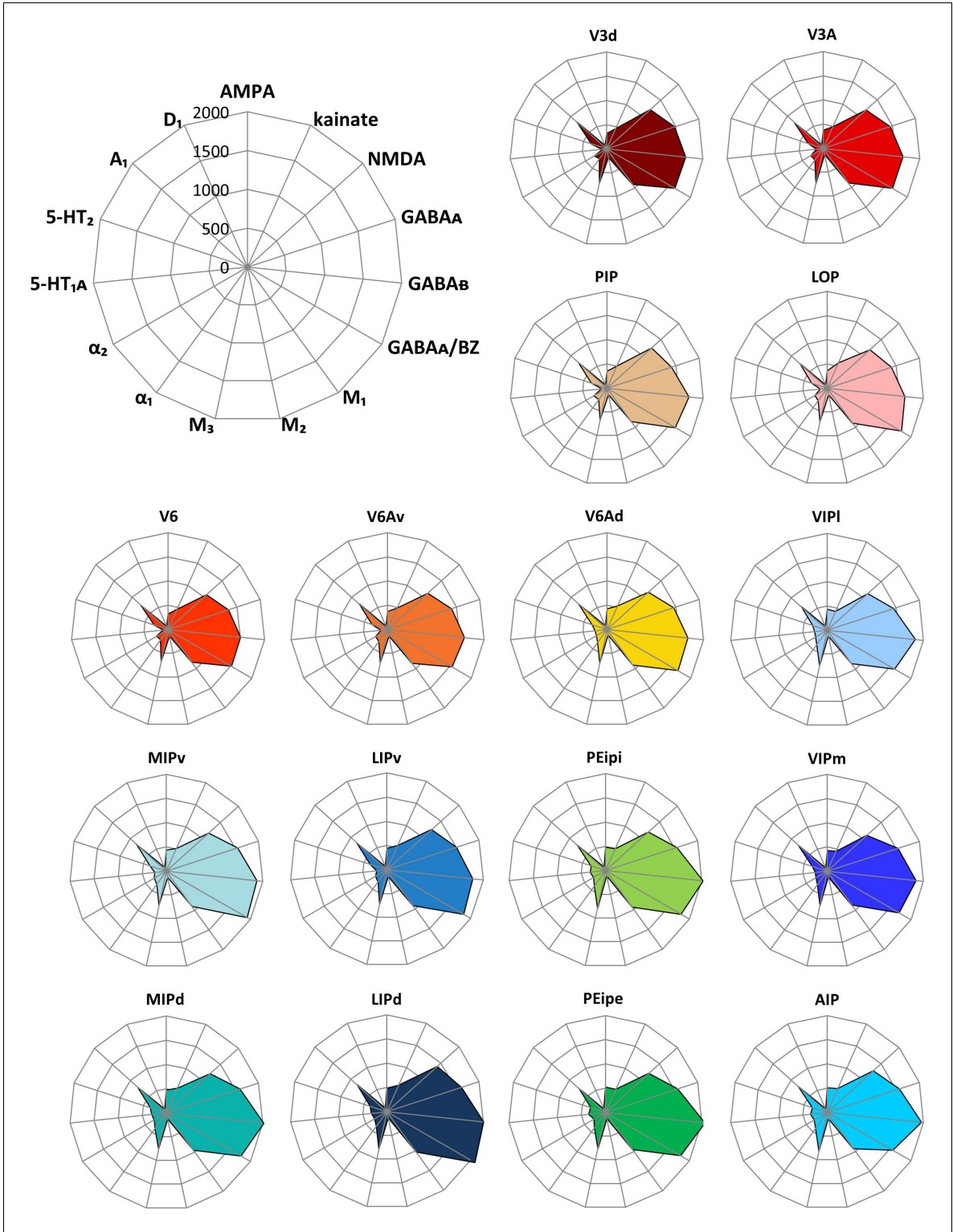

**Figure 11.** Receptor fingerprints of the examined brain areas. Absolute densities in fmol/mg protein of 15 receptors are displayed in polar coordinate plots (scaling 0–2000 fmol/mg protein). For aesthetical reasons, the standard deviations are not displayed in the fingerprints. For mean (and s.d.) densities of each area and receptor type see *Table 1*. The positions of the different receptor types and the axis scaling are identical in all polar plots, and specified in the polar plot at the top left corner of the figure. For abbreviations, see *Figure 2*.

The online version of this article includes the following figure supplement(s) for figure 11:

**Figure supplement 1.** Normalized receptor fingerprints of the examined brain areas.

**Table 1.** Mean receptor densities of IPS areas in fmol/mg protein.

| | | AMPA | Kainate | NMDA | GABA$_A$ | GABA$_B$ | GABA$_A$/BZ | M$_1$ | M$_2$ | M$_3$ | $\alpha_1$ | $\alpha_2$ | 5-HT$_{1A}$ | 5-HT$_2$ | A$_1$ | D$_1$ |
|---|---|---|---|---|---|---|---|---|---|---|---|---|---|---|---|---|
| AIP | mean | 506.5 | 541.0 | 1284.4 | 1564.8 | 1958.2 | 1565.2 | 924.0 | 128.2 | 819.3 | 409.3 | 306.0 | 348.6 | 319.4 | 800.7 | 77.3 |
| | s.d. | 70.1 | 101.0 | 124.5 | 238.1 | 277.9 | 226.9 | 154.5 | 39.3 | 91.8 | 46.0 | 53.3 | 35.2 | 75.9 | 180.4 | 5.1 |
| LIPd | mean | 485.5 | 592.5 | 1397.0 | 1621.0 | 2012.4 | 2097.2 | 1028.2 | 136.7 | 788.6 | 374.2 | 291.7 | 323.3 | 364.9 | 832.3 | 82.0 |
| | s.d. | 67.4 | 60.7 | 199.1 | 159.6 | 290.6 | 350.5 | 186.6 | 41.2 | 113.8 | 28.3 | 25.6 | 50.1 | 73.2 | 183.2 | 7.3 |
| LIPv | mean | 443.2 | 551.0 | 1241.1 | 1508.9 | 1783.3 | 1832.0 | 912.2 | 143.3 | 722.4 | 310.6 | 271.9 | 209.0 | 361.8 | 834.4 | 77.6 |
| | s.d. | 64.8 | 34.3 | 189.3 | 160.3 | 250.3 | 167.5 | 135.7 | 38.0 | 100.9 | 25.5 | 23.7 | 22.2 | 62.8 | 127.6 | 6.4 |
| PEipe | mean | 522.3 | 526.7 | 1203.0 | 1570.2 | 2064.6 | 1792.6 | 943.6 | 134.2 | 766.0 | 390.0 | 278.6 | 355.6 | 323.5 | 733.6 | 81.6 |
| | s.d. | 53.4 | 96.6 | 236.5 | 226.5 | 269.9 | 164.1 | 111.0 | 49.6 | 80.1 | 36.8 | 53.0 | 55.9 | 76.8 | 152.8 | 8.8 |
| PEipi | mean | 483.4 | 495.8 | 1190.1 | 1549.1 | 2024.8 | 1802.3 | 933.4 | 114.6 | 800.2 | 389.2 | 288.0 | 312.3 | 311.2 | 789.8 | 79.2 |
| | s.d. | 85.4 | 95.7 | 135.4 | 194.4 | 286.9 | 114.8 | 77.5 | 40.6 | 108.3 | 39.4 | 53.5 | 50.5 | 55.8 | 111.2 | 7.1 |
| MIPv | mean | 435.0 | 517.2 | 1173.8 | 1549.0 | 1875.7 | 1927.6 | 903.8 | 132.0 | 723.2 | 342.9 | 288.0 | 245.8 | 356.0 | 826.9 | 79.1 |
| | s.d. | 72.9 | 62.9 | 217.1 | 175.1 | 263.9 | 162.7 | 93.7 | 34.9 | 97.2 | 33.7 | 44.9 | 62.6 | 51.8 | 131.0 | 6.7 |
| MIPd | mean | 475.7 | 562.3 | 1210.8 | 1598.3 | 2020.1 | 1779.9 | 951.8 | 126.4 | 760.7 | 406.0 | 291.5 | 307.0 | 350.0 | 796.5 | 85.8 |
| | s.d. | 70.9 | 84.8 | 186.9 | 171.9 | 321.2 | 93.0 | 102.7 | 36.5 | 81.7 | 39.1 | 47.7 | 82.2 | 66.4 | 154.9 | 6.7 |
| VIPl | mean | 431.8 | 423.2 | 1130.1 | 1445.1 | 1829.8 | 1606.6 | 858.7 | 121.6 | 753.2 | 368.3 | 272.5 | 264.3 | 317.7 | 718.4 | 84.3 |
| | s.d. | 75.4 | 91.4 | 151.7 | 243.6 | 203.3 | 142.8 | 81.0 | 41.3 | 76.7 | 27.3 | 49.9 | 28.7 | 69.8 | 120.8 | 9.5 |
| VIPm | mean | 426.7 | 453.9 | 1104.5 | 1534.2 | 1847.5 | 1714.7 | 856.7 | 119.9 | 725.3 | 338.2 | 270.2 | 244.6 | 321.1 | 806.6 | 76.3 |
| | s.d. | 69.3 | 90.1 | 212.8 | 180.7 | 346.4 | 78.8 | 83.7 | 41.3 | 99.9 | 49.7 | 48.5 | 34.3 | 59.0 | 148.1 | 9.1 |
| PIP | mean | 348.6 | 494.9 | 1241.2 | 1400.7 | 1702.0 | 1629.8 | 857.7 | 142.5 | 666.3 | 292.3 | 310.5 | 110.6 | 336.4 | 795.1 | 73.3 |
| | s.d. | 58.7 | 61.3 | 119.2 | 162.5 | 303.9 | 125.5 | 134.0 | 42.3 | 104.3 | 64.7 | 27.0 | 54.6 | 74.8 | 136.8 | 8.3 |
| LOP | mean | 361.6 | 573.1 | 1190.1 | 1396.3 | 1614.2 | 1774.2 | 885.3 | 140.4 | 707.3 | 317.0 | 296.3 | 143.2 | 336.1 | 758.4 | 75.5 |
| | s.d. | 59.0 | 69.3 | 146.2 | 157.7 | 224.2 | 175.3 | 65.5 | 33.3 | 100.3 | 47.6 | 25.1 | 57.1 | 67.3 | 135.1 | 6.0 |
| V3A | mean | 378.0 | 492.2 | 1189.0 | 1455.3 | 1652.9 | 1647.1 | 878.7 | 133.2 | 742.1 | 293.6 | 292.1 | 144.9 | 332.7 | 810.6 | 74.1 |
| | s.d. | 78.5 | 69.4 | 120.1 | 294.1 | 280.1 | 51.5 | 125.0 | 25.4 | 100.9 | 62.0 | 50.2 | 50.3 | 71.0 | 236.9 | 11.9 |
| V3d | mean | 317.9 | 452.0 | 1213.6 | 1476.8 | 1644.7 | 1627.7 | 906.6 | 178.9 | 727.6 | 249.6 | 294.6 | 87.9 | 331.0 | 827.0 | 74.9 |
| | s.d. | 66.2 | 82.4 | 129.3 | 405.2 | 289.8 | 120.2 | 99.0 | 60.5 | 93.5 | 45.8 | 37.8 | 37.3 | 74.0 | 192.6 | 7.0 |
| V6Ad | mean | 427.2 | 536.3 | 1148.7 | 1450.0 | 1679.1 | 1698.1 | 904.3 | 118.3 | 710.1 | 329.1 | 248.6 | 245.9 | 323.2 | 782.7 | 80.4 |
| | s.d. | 71.1 | 87.4 | 116.5 | 265.0 | 335.4 | 141.4 | 74.3 | 36.7 | 92.9 | 35.5 | 36.0 | 46.3 | 68.8 | 147.4 | 8.1 |
| V6Av | mean | 373.8 | 483.1 | 1115.6 | 1393.7 | 1595.7 | 1543.9 | 857.0 | 134.3 | 706.8 | 305.4 | 286.3 | 136.9 | 336.9 | 790.1 | 76.0 |
| | s.d. | 71.3 | 59.5 | 71.1 | 193.3 | 277.0 | 116.1 | 64.7 | 32.7 | 91.5 | 29.3 | 23.2 | 38.7 | 52.5 | 106.2 | 4.6 |
| V6 | mean | 319.2 | 457.8 | 1073.0 | 1317.3 | 1505.0 | 1514.6 | 832.1 | 136.7 | 675.9 | 260.1 | 274.9 | 99.8 | 301.8 | 754.1 | 71.5 |
| | s.d. | 57.7 | 58.9 | 142.3 | 233.8 | 397.6 | 34.2 | 66.2 | 40.6 | 117.0 | 47.5 | 37.1 | 39.4 | 73.5 | 129.2 | 6.7 |

We largely confirmed the delineations on the medial bank of the IPS where it joins the POS (*Impieri et al., 2019*; *Luppino et al., 2005*; *Zeki, 1986*; *Galletti et al., 1996*; *Galletti et al., 1999a*; *Galletti et al., 1999b*). Two functionally distinct areas were originally detected in this region: a ventral, purely visual area V6, and a dorsal area V6A (*Zeki, 1986*; *Galletti et al., 1996*). Luppino, Hamed (*Luppino et al., 2005*) further separated V6A into two areas (V6Av and V6Ad) by using myeloarchitecture combined with the distribution of SMI-32 immunoreactivity. In the present study and in that by Impieri, Zilles (*Impieri et al., 2019*), multiple receptor architecture corroborated the subdivisions of area V6A, as well as the location and extent of area V6. Differences between V6Ad, V6Av and V6 are prominently indicated by the AMPA, NMDA, M$_1$, M$_2$, M$_3$ and $\alpha_1$ receptors. Interestingly, for all receptors mentioned above (except for M$_2$), we found that the dorsalmost area V6Ad presents a higher receptor density pattern compared to the V6Av, and the ventralmost area V6 always shows the lowest densities. This may be explained by their functional heterogeneity, which will be discussed in the next part.

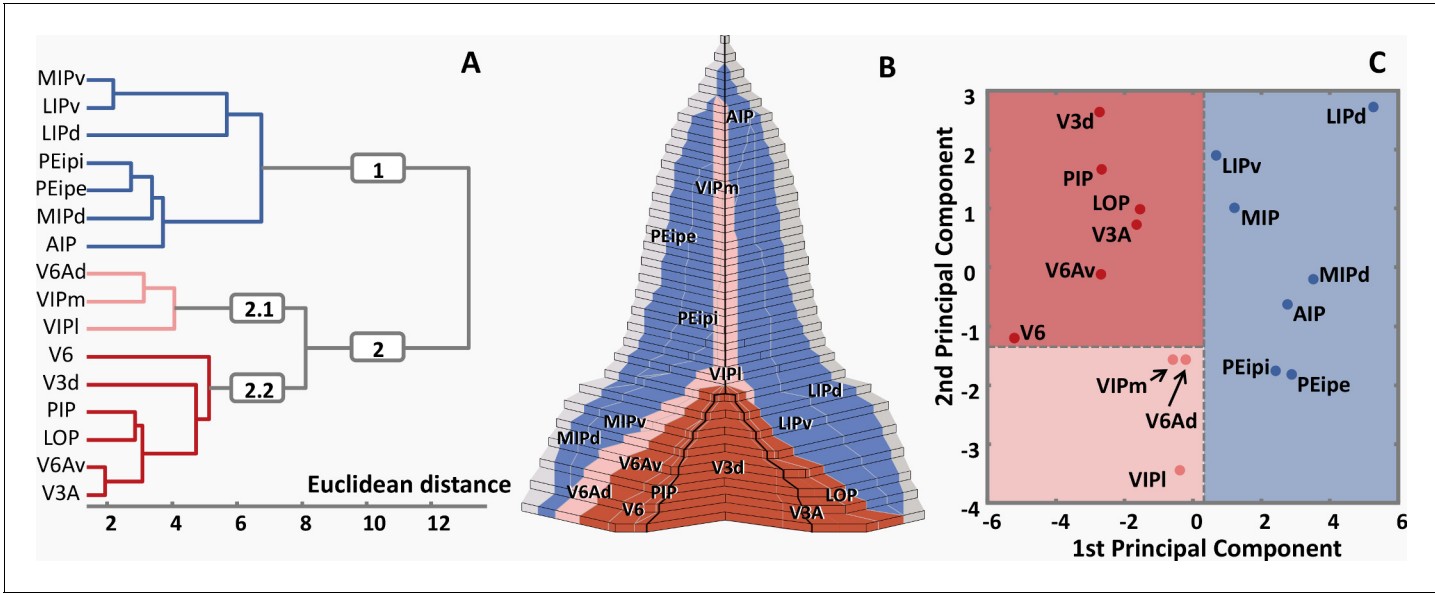

**Figure 12.** Receptor-driven clustering of the IPS subdivisions. (A) Hierarchical cluster analysis reveals 3 receptor-architectonically distinct clusters: a caudal cluster with visual areas (blue); an intermediate group of areas VIPm, VIPl and V6Ad (pink); and a rostral group consisting of all areas located on the bilateral walls of IPS (red). (B) Three clusters are displayed in the 2D flat map, same color coding as in (A). (C) Principal component analysis. The distances between areas represent the Eigenvalues of the first and second principal components, three clusters are clearly segregated by the first and second principal component, same color coding as in (A) and (B). For abbreviations, see *Figure 2*.

## Areas located in the IPS

We were able to identify nine cyto- and receptor architectonically distinct areas completely buried in the IPS of the macaque brain. Areas MIPd and MIPv cover the posterior portion of the medial wall, and areas PEipe and PEipi occupy the rostralmost portion of its medial bank. On the lateral bank, the posterior two-thirds contain two architectonically distinct subdivisions of LIP: LIPd and LIPv. The fundus of the IPS comprises medial and lateral subdivisions of VIP, that is, VIPm and VIPl. The anterior part of the IPS fundus is occupied by areas AIP and the rostral portion of VIP.

There are discrepancies concerning the number of areas found in the fundus of the rostral portion of the IPS. Whereas *Preuss and Goldman-Rakic, 1991* described a single area, namely VIP, *Lewis and Van Essen, 2000a* found evidence for consistent cytoarchitectonic differences between medial and lateral subdivisions of VIP. We were able to confirm the existence of these two subdivisions both by the cyto-/myelo- and receptor architectonic analyses. Similarly to the description of *Lewis and Van Essen, 2000a*, we found VIPm to have more densely packed cells than VIPl, particularly in layers III to V, and the darkly stained pyramids in layer III are larger and more prominent in VIPl than in VIPm. Furthermore, differences in the mean densities of the AMPA, GABAergic, $M_2$, $\alpha_1$, $5\text{-HT}_{1A}$, $A_1$ and $D_1$ receptors also revealed the border between these two areas. Thereby, VIPl presented higher densities of $M_2$, $5\text{-HT}_{1A}$, $A_1$, $D_1$ receptors and lower densities of AMPA, GABAergic, $\alpha_1$ receptors than did VIPm.

On the lateral bank, area LIP (or POa) was identified by most previous schemes (*Andersen et al., 1985*; *Seltzer and Pandya, 1980*; *Preuss and Goldman-Rakic, 1991*; *Lewis and Van Essen, 2000a*; *Blatt et al., 1990*). It has been further subdivided into dorsal (LIPd) and ventral subdivisions (LIPv), or external (POa-e) and internal (POa-i) subdivisions respectively, based on both cyto- (*Lewis and Van Essen, 2000a*) and myeloarchitecture (*Preuss and Goldman-Rakic, 1991*; *Blatt et al., 1990*). Our present scheme largely confirms the map published by *Lewis and Van Essen, 2000a*, who recognized dorsal and ventral subdivisions of LIP based on histochemical and immunohistochemical stains. As compared with area LIPd, LIPv contained a more densely packed layer III with larger pyramidal cells, but a less densely packed layer V with smaller cells in cytoarchitecture. Additionally, LIPv could be distinguished from area LIPd by its lower mean densities of almost all receptors.

Although the organization of the medial wall of the IPS has been the focus of numerous studies (*Lewis and Van Essen, 2000a*; *Bakola et al., 2017*; *Seelke et al., 2012*), a consensus on the number

and arrangement of areas has emerged for only a modest fraction of its total extent. Besides the terminological roots, previous parcellation schemes also differ considerably in terms of the exact number and precise arrangement of distinct areas, such as their size, shape, and location relative to macroanatomic landmarks. According to the studies from *Lewis and Van Essen, 2000a* and Bakola, Passarelli (*Bakola et al., 2017*), three areas occupied the medial wall of the IPS: area PO is located in the junction with POS, and the remaining rostral part consists of two distinct areas. The middle area was named MIP (*Lewis and Van Essen, 2000a*; *Bakola et al., 2017*), whereas the rostralmost area was identified as area 5V (*Lewis and Van Essen, 2000a*) or as area PEip (*Bakola et al., 2017*; *Gamberini et al., 2020a*). The present partitioning scheme was able to confirm the existence of these two areas, their location and extent are more similar to the scheme charted by *Bakola et al., 2017*, so we followed their nomenclature. Dorsal to the originally defined area MIP, we identified an architectonic distinct area as a newly described subdivision, which was originally detected by connection studies (*Bakola et al., 2010*; *Bakola et al., 2013*). This new subdivision could be distinguished from the original MIP due to its more densely packed cells, especially in layers II, IIIc and VIb. So, we renamed area MIP as MIPv, and named this new subdivision as MIPd. At the same time, evidence from receptor architecture also indicates the existence of MIPd, the differences in receptor densities between cortical layers are more apparent in MIPv than in MIPd. Additionally, for kainate, $M_2$, and $\alpha_1$ receptors, area MIPv has the lowest densities in layers IIIc and V, thus resulting in a sharply definable boundary with MIPd. Area PEip has previously been described as architectonically homogenous (*Bakola et al., 2017*; *Seelke et al., 2012*); however, we found evidence for consistent architectonic differences between exteral and interal subdivisions, PEipe and PEipi. In cytoarchitecture, PEipe shows a clearer lamination and a higher cell packing density than PEipi. In receptor architecture, the main difference between PEipe and PEipi occurred in the supragranular layers. For layers I-IIIb, most of receptors ($\alpha_1$, $GABA_B$, $M_1$, $M_3$ and $5\text{-}HT_{1A}$ receptors) present higher densities in area PEipe, whereas, for $M_1$ receptors and $GABA_A$/BZ binding sites, the densities of PEipi are higher than those of PEipe. However, their functional heterogeneity remains to be determined.

## Molecular and functional organization of IPS

One of the purposes of this study was to analyze the organization of the IPS with a focus on the relationship between its molecular and functional segregation since similarities in fingerprints have been postulated to be indicative of the existence of a network of cortical areas characterized by their comparable receptor expression and participation in the same functional system (*Palomero-Gallagher and Zilles, 2018*; *Zilles and Amunts, 2009*; *Zilles et al., 2015*; *Palomero-Gallagher et al., 2009*). Furthermore, since the function of a cortical area requires a well-tuned receptor balance, the receptor fingerprint of each area may represent an index of its physiological, connectional and functional properties (*Palomero-Gallagher and Zilles, 2018*; *Zilles and Amunts, 2009*; *Zilles et al., 2002a*; *Palomero-Gallagher and Zilles, 2019b*; *Rakic et al., 1988*).

There is a gradual direction trend in the functional organization of the primate IPS: the anterior portion is more concerned with sensorimotor processing, whereas the posterior part is involved in somatosensory and visual processing (*Gamberini et al., 2015*; *Felleman and Van Essen, 1991*; *Heekeren et al., 2008*). Furthermore, with respect to the walls of the IPS, areas located on the medial wall are more active during arm movements (*Colby and Duhamel, 1991*; *Grefkes and Fink, 2005*; *Seelke et al., 2012*), and areas located on the lateral wall are more associated with eye movements (*Blatt et al., 1990*; *Patel et al., 2010*). As revealed by the cluster analyses of the receptor fingerprints of areas examined in the present study, three distinct clusters with a grouping of areas with distinct macroanatomical locations along the rostro-caudal axis of the hemisphere were observed:

## Areas of the caudal cluster

The caudal cluster includes areas V3d, V3A, PIP, LOP, V6 and V6Av. Area V3d is the dorsal portion of area V3, which together with V3A, constitute part of the extrastriate early visual cortex (*Felleman and Van Essen, 1991*; *Felleman et al., 1997*), and receive direct input from V1 (*Gattass et al., 1988*; *Felleman et al., 1997*). These regions respond differently to shape, color, direction and motion of objects (*Livingstone and Hubel, 1988*; *Zeki, 1993*). PIP is also a visual area with motion sensitivity (*Colby et al., 1988*; *Vanduffel et al., 2001*), which probably belongs to the

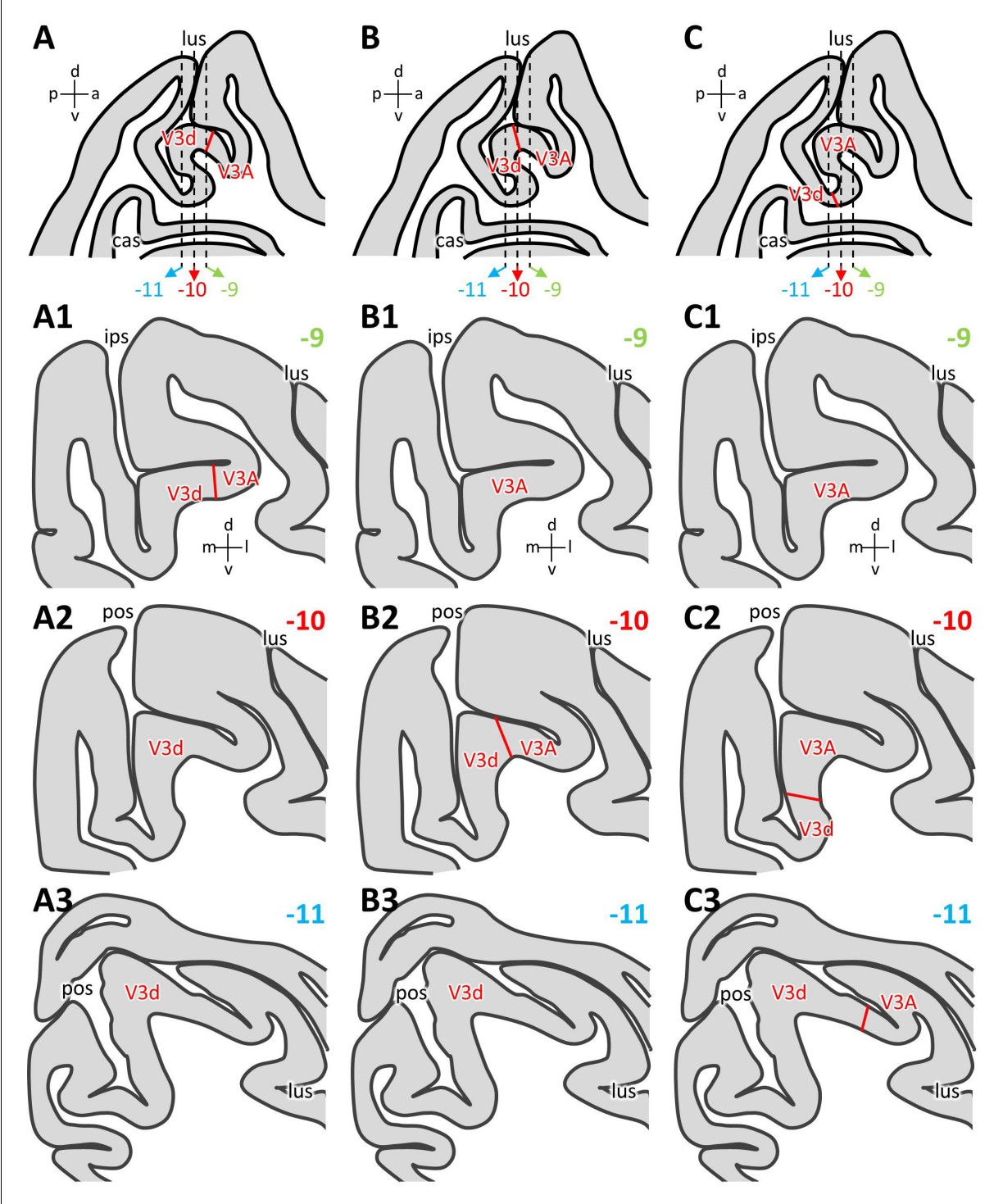

**Figure 13.** Inter-individual variability in the position of the V3d/V3A border. Schematic representation of an exemplary parasagittal section located 10 mm to the right of the midline, on which variations in the position of the V3d/V3A border in relation to the anectant gyrus are illustrated (A, B, C). Dashed lines indicate the rostro-caudal position of the coronal sections shown below and which are located 9 mm (A1, B1, C1), 10 mm (A2, B2, C2) and 11 mm (A3, B3, C3) caudal to the interaural surface (*Saleem and Logothetis, 2012*). (A1–A3) Schematic representation of areas found on the anectant gyrus when the V3d/V3A border was detected on its rostral wall (A). (B1–B3) Schematic representation of areas found on the anectant gyrus when the V3d/V3A border was detected on its apex (B). (C1–C3) Schematic representation of areas found on the anectant gyrus when the V3d/V3A border was detected on its caudal wall (C). Abbreviations: *a* anterior, *cas* calcarine sulcus, *d* dorsal, *ips* intraparietal sulcus, *l* lateral, *lus* lunate sulcus, *m* medial, *p* posterior, *pos* parieto-occipital sulcus, *v* ventral.

neural circuits involved in visual-guided grasping in macaque monkeys (*Jeannerod et al., 1994*). Our area LOP overlaps substantially with the functionally identified caudal intraparietal area (CIP), which is involved in the analysis of 3D object features (shape, orientation and curvatures; *Katsuyama et al., 2010*; *Alizadeh et al., 2018*). Area V6 is sensitive to the direction, orientation, and speed of motion (*Galletti et al., 1996*) and to the movement of objects (*Galletti and Fattori, 2003*). It is a retinotopically organized unimodal area; that is, it contains only cells responsive to visual stimuli (*Gamberini et al., 2011*). Area V6Av is not retinotopically organized, and although most of its cells are responsive to visual stimuli, twenty percent are also activated by somatosensory stimuli (*Gamberini et al., 2011*). These findings suggest that V6Av is primarily a visual area that provides the visuo-motor integration network with visual information (*Gamberini et al., 2009*; *Passarelli et al., 2011*; *Gamberini et al., 2020b*). Viewed as a whole, this caudal cluster mainly encompasses areas associated with the processing of visual information, although V6Av also participates in visuomotor coordination.

## Areas of the intermediate cluster

The intermediate cluster consists of areas V6Ad, VIPm and VIPl, and may be interpreted as a multi-modal integration cluster serving as an interface between the somatosensory and visual systems. V6Ad is a bimodal visual-somatic area, which is likely involved in the visual guidance of arm movement (*Galletti et al., 1997*) and object grasping (*Caminiti et al., 1991*; *Raos et al., 2004*). Multi-modal area V6Ad receives input from many parietal and frontal areas, as well as from visual areas, though to a lesser extent (*Boussaoud et al., 1998*; *Gregoriou et al., 2005*). Functionally, V6Ad is more involved in the control of visually guided grasp actions (*Fattori et al., 2010*), whereas V6Av (which is located in the caudal cluster) is more involved in the visual analysis of these actions (*Gamberini et al., 2011*; *Fattori et al., 2010*). Thus, V6Av has been classified as a primarily visual area, whereas V6Ad is considered a truly multimodal area that integrates both visual and somatosensory inputs. This functional distinction is reflected by differences in the receptor architecture of V6Ad and V6Av, since they are assigned to different clusters by the multimodal cluster analyses.

Area VIP is a complex region which receives projections not only from several visual areas, but also from somatosensory, auditory, and motor areas as well as from some poly-sensory cortices (*Colby and Duhamel, 1991*; *Lewis and Van Essen, 2000a*; *Shao et al., 2018*). That means, VIP is a polymodal association region responding to visual, tactile, and auditory stimuli. Furthermore, a previous tracer study (*Lewis and Van Essen, 2000b*) indicates that VIPl tends to have stronger connections with visual-related areas, whereas VIPm is more strongly connected with sensorimotor-related areas. Functionally, VIP participates in optic flow processing, quantity-processing, auditory processing and spatial attention additionally (*Bremmer et al., 2002*; *Harvey et al., 2017*; *Klam and Graf, 2003*). However, given the limited number of corresponding research, the functional heterogeneity between VIPl and VIPm will need to be resolved by future studies.

In general, the intermediate cluster is important for multisensory integration, particularly for visual and somatosensory input.

## Areas of the rostral cluster

The rostral cluster encompasses area AIP mainly in the fundus of the IPS, areas LIPd and LIPv on the lateral wall, as well as areas PEipi and PEipe on the rostral portion of the medial wall, and areas MIPd, and MIPv on the caudo-medial part of the IPS. These areas are involved in sensorimotor processing and are particularly important for the coordination of eye and upper limb movements (*Blatt et al., 1990*; *Lewis and Van Essen, 2000b*; *Colby and Goldberg, 1999*).

Interestingly, there is a medio-lateral functional gradient among these areas. Neurons located on the medial wall of the IPS are more involved in arm movements, whereas neurons situated in the lateral bank are predominantly responsive to eye movements (*Grefkes and Fink, 2005*; *Patel et al., 2010*; *Colby and Goldberg, 1999*). Specifically, MIP is mainly responsible for mediating planning, execution and monitoring of reaching movements (*Bakola et al., 2017*; *Vingerhoets, 2014*), and is also considered to be part of a neural network that detects movement errors or corrects movements (*Desmurget and Grafton, 2000*; *Kalaska et al., 2003*). Until now, just a few studies (*Bakola et al., 2010*; *Bakola et al., 2013*) reported on a functional heterogeneity within MIP. Based on their results, we suppose MIPd is more active during somatosensory process, because the cells responsive to

sensory stimuli were mainly found in MIPd instead of in MIPv. Correspondingly, MIPv contains bimodal cells (both visual and somatosensory), and the visual responsivity increases with depth along the medial wall. Conversely, the parietal eye field LIP, located on the lateral bank of the sulcus, is specifically involved in a neural network for controlling eye movements and attention (*Blatt et al., 1990*; *Patel et al., 2010*). Based on the results from previous electrophysiological and functional studies (*Blatt et al., 1990*; *Lewis and Van Essen, 2000b*; *Liu et al., 2010*), dorsal LIP (LIPd) representing central visual fields, has more connections with 'featural' processing pathways in inferotemporal cortex. Thus, LIPd is primarily involved in oculomotor processes. In contrast, ventral LIP (LIPv) contributes to both attentional and oculomotor processes. The connection pattern of LIPv confirms that it represents peripheral visual fields, since it is mainly connected with areas of motion processing pathways.

A clear rostro-caudal functional gradient is found along the medial wall of the IPS. Although PEip and MIP are both somatosensory-related areas and have complementary roles in sensorimotor behavior (*Bakola et al., 2010*; *Seelke et al., 2012*; *Rathelot et al., 2017*), PEip is more directly involved in the control of limb movement in comparison with MIP (*Bakola et al., 2017*; *Bakola et al., 2010*), and PEip neurons can execute these movements without visual information (*Evangeliou et al., 2009*; *Nelissen and Vanduffel, 2011*). Thus, areas within the anterior part of the IPS are more active during sensorimotor processing, whereas areas in the posterior portion are more active during visuo-somatic processing.

Areas encompassed by the anterior cluster constitute a site of convergent visual and somatic-related information, and are directly involved in movement and decision-related processes in arm and eye movements. Furthermore, this functional heterogeneity is subserved by receptor architectonically distinct areas.

Interestingly, this shift in functional systems, from a caudo-lateral visual domain to a rostro-medial sensorimotor domain (*Grefkes and Fink, 2005*; *Lewis and Van Essen, 2000b*; *Gamberini et al., 2020a*; *Guipponi et al., 2013*), is also the most significant trend reflected by changes in the shape and/or size of the fingerprints. From an overall perspective, the rostralmost areas (i.e. PEipe, AIP) and the caudalmost areas (i.e. V3d, V6) show the most obvious dissimilarity regarding shape and size of their fingerprints, whereas other areas display a gradual change in their shape and size. Additionally, sensorimotor responsivity decreases with depth along the both walls of IPS (*Blatt et al., 1990*; *Bakola et al., 2010*; *Bakola et al., 2013*; *Liu et al., 2010*): multimodal integration appears to be stronger in ventral subdivisions (i.e. areas PEipi and LIPv) near the fundus of the IPS than in their dorsally paired subdivisions (i.e. areas PEipe and LIPd). This trend can be associated with dorso-ventral variations in receptor fingerprints: PEipe and LIPd contained higher densities, particularly of GABA$_B$ receptors and GABA$_A$/BZ-binding sites, than did PEipi and LIPv, respectively.

The present study provides a comprehensive parcellation scheme of the macaque IPS and its junction with POS with explicit information on the position of the borders of all examined areas relative to macroanatomical landmarks. The multimodal, quantitative, and statistically testable approach not only confirmed the existence and extent of some previously described but controversially discussed areas, but also enabled identification of novel subdivisions (dorsal and ventral subdivisions of area MIP and internal and external parts of area PEip). The receptor fingerprints reveal the molecular organization of the macaque IPS and provide insights into the close relationship between the structural and functional segregation of this brain region in the macaque monkey brain.

## Materials and methods

All experimental protocols were carried out in accordance with the guidelines of the European Communities Council Directive for the care and use of animals for scientific purposes.

### Postmortem tissue and histological procedures

Two samples of postmortem macaque brains were used in the study.

Sample 1 consisted of four hemispheres from three Macaca fascicularis monkeys (animal ID #11539: left and right hemispheres, #11543: left hemisphere, #11530: left hemisphere) obtained from Covance Laboratories (Münster, Germany), which were used for cyto-, myelo- and receptor architectonic analyses. All three specimens were male and between 6 and 8 years old. The animals were sacrificed by means of a lethal i.v. injection of sodium pentobarbital, and the brains removed

immediately. Each hemisphere was cut coronally into an anterior and a posterior slab at the height of the most caudal portion of the central sulcus. Since fixation of the brain modifies the structure of receptor proteins and leads to changes in specific and nonspecific binding (*Palomero-Gallagher and Zilles, 2018*), we used only unfixed, deep frozen brains for receptor autoradiography. After shock freezing of the tissue at $-40°C$ for 10–15 min in liquid isopentane, the slabs were stored at $-80°C$ in airtight plastic bags. Each brain slab was cut into serial coronal sections (20 µm thickness) at $-20°C$ using a cryostat microtome (CM 3050, Leica, Germany). The sections were thaw-mounted onto glass slides, and freeze-dried overnight. Adjacent sections were processed for the visualization of cell bodies, myelin or quantitative in vitro receptor autoradiography as described below.

Sample 2 consisted of the brain of a Macaca mulatta (animal ID: DP1) monkey obtained from Deepak N. Pandya and used for cytoarchitectonic analysis. The monkey was deeply anesthetized with sodium pentobarbital, transcardially perfused with cold saline followed by 10% buffered forma-lin and the brain was stored in a buffered formalin solution for several months. After a dehydration step in graded alcohols (70% to 100% propanol), the brain was embedded in paraffin and serially sectioned at 20 µm in the coronal plane. Every fifth section was mounted on gelatin coated slides, stained for cell bodies with a modified silver cell-body staining (*Merker, 1983*) that provides a high contrast between cell bodies and neuropil.

## Receptor binding procedure

For sample 1, neighboring sections were incubated with a tritiated receptor ligand or stained for cell bodies (*Merker, 1983*) or myelin (*Gallyas, 1979*). Thus, a group of sections obtained from the same rostro-caudal sectioning level included information concerning receptor distribution patterns, cyto- and myeloarchitecture. In the present study, we investigated binding sites of fifteen different receptors from various neurotransmitter systems: glutamatergic (AMPA, kainate, NMDA), GABAergic (GABA$_A$, GABA$_B$, GABA$_A$ associated benzodiazepine [GABA$_A$/BZ] binding sites), cholinergic (muscarinic M$_1$, M$_2$, M$_3$), adrenergic ($\alpha_1$, $\alpha_2$), serotoninergic (5-HT$_{1A}$, 5-HT$_2$), dopaminergic (D$_1$), and adenosinergic (A$_1$).

Detailed information of the autoradiographical labelling method was described in previous publications (*Palomero-Gallagher and Zilles, 2018*; *Zilles et al., 2002a*). In short, it consists of three steps: 1) preincubation for rehydration of sections and removal of endogenous ligands that could block the binding; 2) main incubation with tritiated ligands, and 3) rinsing step to stop the binding procedure and remove unbound radioactive ligand. During the main incubation step, most sections were incubated in a buffer solution containing the tritiated ligand to demonstrate total binding, and selected sections were incubated in another buffer solution containing the tritiated ligand with an appropriate nonlabelled displacer to identify nonspecific binding. For all receptors in the present study, nonspecific binding was less than 5% of the total binding. Thus, the total binding of each receptor could be accepted as an estimate of the specific binding. Detailed incubation protocols are summarized in *Table 2*.

After air-drying, the sections were exposed together with plastic scales with known concentrations of radioactivity to films sensitive to β -radiation (Hyperfilm, Amersham) for 4–18 weeks depending on the ligand used.

For further details on image acquisition see Zilles, Schleicher (*Zilles et al., 2002b*) and Palomero-Gallagher and Zilles (*Palomero-Gallagher and Zilles, 2018*).

## Analysis of cytoarchitecture

Histological sections processed for the visualization of cell bodies or of myelin were scanned at an in-plane resolution of 1 µm per pixel using a light microscope (Axioplan 2 imaging, Zeiss, Germany) equipped with a motor-operated stage controlled by the KS400 (Zeiss, Germany) image-analyzing system (version 3.0) and Axiovision (version 4.6).

The position of borders detected by visual inspection was confirmed using a quantitative and statistically testable approach. This involved the use of MatLab (The MathWorks, Natick, MA) based in-house scripts for the computation of gray level index (GLI) images (*Palomero-Gallagher and Zilles, 2018*; *Schleicher et al., 1999*) from a sector of the digitized histological section. Each pixel in the GLI image represents the local volume fraction of cell bodies in the corresponding measuring field (*Wree et al., 1982*; *Schleicher and Zilles, 1990*).

**Table 2.** Incubation protocols.

| Transmitter | Receptor | Ligand (nM) | Property | Displacer | Incubation buffer | Pre-incubation | Main incubation | Final rinsing |
|---|---|---|---|---|---|---|---|---|
| Glutamate | AMPA | [³H]-AMPA (10.0) | Ag | Quisqualate (10 µM) | 50 mM Tris-acetate (pH 7.2) [+ 100 mM KSCN]* | 3x10 min, 4°C | 45 min, 4°C | 1) 4x4 sec 2) Acetone/ glutaraldehyde (100 ml + 2,5 ml), 2x2 sec, 4°C |
| | NMDA | [³H]-MK-801 (3.3) | Ant | (+)MK-801 (100 µM) | 50 mM Tris-acetate (pH 7.2) + 50 µM glutmate [+ 30 µM glycine + 50 µM spermidine]* | 15 min, 4°C | 60 min, 22°C | 1) 2x5 min, 4°C 2) Distilled water, 1x22°C |
| | Kainate | [³H]-Kainate (9.4) | Ag | SYM 2081 (100 µM) | 50 mM Tris-acetate (pH 7.2) [+ 10 mM Ca²⁺-acetate]* | 3x10 min, 4°C | 45 min, 4°C | 1) 3x4 sec 2) Acetone/ glutaraldehyde (100 ml + 2,5 ml), 2x2 sec, 22°C |
| GABA | GABA_A | [³H]-Muscimol (7.7) | Ag | GABA (10 µM) | 50 mM Tris-citrate (pH 7.0) | 3x5 min, 4°C | 40 min, 4°C | 1) 3x3 sec, 4°C 2) Distilled water, 1x22°C |
| | GABA_B | [³H]-CGP 54626 (2.0) | Ant | CGP 55845 (100 µM) | 50 mM Tris-HCl (pH 7.2) + 2.5 mM CaCl₂ | 3x5 min, 4°C | 60 min, 4°C | 1) 3x2 sec, 4°C 2) Distilled water, 1x22°C |
| | GABA_A/ BZ | [³H]-Flumazenil (1.0) | Ant | Clonazepam (2 µM) | 170 mM Tris-HCl (pH 7.4) | 15 min, 4°C | 60 min, 4°C | 1) 2x1 min, 4°C 2) Distilled water, 1x22°C |
| Acetylcholine | M₁ | [³H]-Pirenzepine (1.0) | Ant | Pirenzepine (2 µM) | Modified Krebs buffer (pH 7.4) | 15 min, 4°C | 60 min, 4°C | 1) 2x1 min, 4°C 2) Distilled water, 1x22°C |
| | M₂ | [³H]-Oxotremorine-M (1.7) | Ag | Carbachol (10 µM) | 20 mM HEPES-Tris (pH 7.5) + 10 mM MgCl₂ + 300 nM Pirenzepine | 20 min, 22°C | 60 min, 22°C | 1) 2x2 min, 4°C 2) Distilled water, 1x22°C |
| | M₃ | [³H]-4-DAMP (1.0) | Ant | Atropine sulfate (10 µM) | 50 mM Tris-HCl (pH 7.4) + 0.1 mM PSMF + 1mM EDTA | 15 min, 22° C | 45 min, 22° C | 1) 2x5 min, 4° C 2) distilled water, 1x22°C |
| Noradrenaline | α₁ | [³H]-Prazosin (0.2) | Ant | Phentolamine Mesylate (10 µM) | 50 mM Na/K-phosphate buffer (pH 7.4) | 15 min, 22°C | 60 min, 22°C | 1) 2x5 min, 4°C 2) Distilled water, 1x22°C |
| | α₂ | [³H]-UK 14,304 (0,64) | Ag | Phentolamine Mesylate (10 µM) | 50 mM Tris-HCl + 100 µM MnCl₂ (pH 7.7) | 15 min, 22°C | 90 min, 22°C | 1) 5 min, 4°C 2) Distilled water, 1x22°C |
| Serotonin | 5-HT_1A | [³H]-8-OH-DPAT (1.0) | Ag | 5-Hydroxy-tryptamine (1 µM) | 170 mM Tris-HCl (pH 7.4) [+ 4 mM CaCl₂ + 0.01% ascorbate]* | 30 min, 22°C | 60 min, 22°C | 1) 5 min, 4°C 2) Distilled water, 3x22°C |
| | 5-HT₂ | [³H]-Ketanserin (1.14) | Ant | Mianserin (10 µM) | 170 mM Tris-HCl (pH 7.7) | 30 min, 22°C | 120 min, 22°C | 1) 2x10 min, 4°C 2) Distilled water, 3x22°C |
| Dopamine | D₁ | [³H]-SCH 23390 (1.67) | Ant | SKF 83566 (1 µM) | 50 mM Tris-HCl + 120 mM NaCl + 5 mM KCl + 2 mM CaCl₂ + 1 mM MgCl₂ (pH 7.4) | 20 min, 22°C | 90 min, 22°C | 1) 2x20 min, 4°C 2) Distilled water, 1x22°C |
| Adenosine | A₁ | [³H]-DPCPX (1.0) | Ant | R-PIA (100 µM) | 170 mM Tris-HCl + 2 Units/I Adenosine deaminase [+ 100 µM Gpp(NH)p]* (pH 7.4) | 15 min, 4°C | 120 min, 22°C | 1) 2x5 min, 4°C 2) Distilled water, 1x22°C |

* Only included in the main incubation. Ag agonist, Ant antagonist

The quantitative approach to identification of cortical borders is based on the concept that each area has a laminar pattern which differs from that of adjacent areas. In order to quantify the laminar GLI value changes from the pial surface to the border between gray and white matter, equidistant GLI profiles were extracted vertically to the cortical surface (*Figures 5B* and *7B*; *Palomero-*

*Gallagher and Zilles, 2018*; *Schleicher et al., 2005*). To compensate for variations in cortical thickness, the length of each profile was normalized using linear interposition to a cortical thickness of 100% (0% = pial surface; 100% = border to white matter). Each normalized GLI profile can be parametrized to quantitatively describe the laminar distribution of the volume fraction of the cell bodies, here, we extracted five features (mean GLI value, mean of cortical depth, standard deviation of the mean GLI, skewness, kurtosis) from the original GLI profile and the corresponding five features from its differential quotient. Thus, these 10 parameters constitute a feature vector which can be used to measure the degree of similarity or dissimilarity in the shape of profiles as a Mahalanobis distance (MD; *Mahalanobis et al., 1949*).

Maximum differences in profile shape can be expected at the interface between the feature vectors of two groups (or blocks) of profiles which represent adjacent cortical areas. Thus a MD function was determined by computing the distances between all pairs of neighboring blocks of profiles and plotting the values as a function of the block's position. To control the stability of the distance function depending on the block size (number of profiles in a block), this procedure was repeated for varying block sizes (10–24 adjacent profiles; *Figures 5C* and *7C*). A maximum of the MD function indicated a potential cytoarchitectonical border and was accepted as such after confirmation of its statistical significance by means of a Hotelling's T2 test (p<0.01) and Bonferroni correction (*Palomero-Gallagher and Zilles, 2018*; *Schleicher et al., 2005*). Furthermore, since the presence of blood vessels can at times influence the MD, the biological relevance of significant maxima was confirmed by microscopic inspection of the histological section and by comparing spatially corresponding significant maxima from adjoining sections. Since the probability of locating such maxima at comparable sites in a stack of sections by reasons other than changes in cortical organization is exceedingly small, this procedure excluded biologically meaningless maxima which may be caused by artifacts (e.g. ruptures, folds) or local inhomogeneities in microstructure such as blood vessels or atypical cell clusters (*Palomero-Gallagher and Zilles, 2018*; *Schleicher et al., 2005*).

## Analysis of receptor architecture

Autoradiographs were digitized using a video-based image analyzing system composed of a CCD-camera (Axiocam MRm, Zeiss, Germany) and the image processing software Axiovision (Zeiss, Germany), as well as in-house-developed Matlab (The MathWorks, Natrick, MA) scripts. The resulting eight-bit images represent the regional and laminar distribution of receptor-binding sites. The plastic scales were used to compute a transformation curve indicating the relationship between gray values of an autoradiograph and concentrations of radioactivity in the tissue (linearized autoradiographs), so that the gray value of each pixel in the image could then, with the information concerning specific experimental conditions, be converted into a binding site concentration in fmol/mg protein. To provide a clear visualization of the regional and laminar receptor distribution patterns, digitized autoradiographs were linearly contrast enhanced and pseudo-color coded.

To ensure that the cytoarchitectonically defined areas were identified in a comparable way by their receptor distribution patterns, borders between the IPS areas were delineated within all receptor autoradiographs by means of the same approach as described above, but profiles were extracted from the linearized autoradiographs (*Figures 6* and *8*; *Schleicher et al., 2005*). These receptor profiles were also used to quantify receptor densities in the identified areas. To this purpose, for each area, hemisphere and receptor, 90–150 receptor profiles (extracted from three to five sections) were averaged to obtain mean areal receptor densities (i.e. averaged over all cortical layers; *Palomero-Gallagher and Zilles, 2018*; *Zilles et al., 2002b*; *Schleicher et al., 2005*).

In order to identify significant differences in receptor densities between adjacent cortical areas, we performed a series of linear mixed-effects models using the R programming language (version: 3.6.3; *R Development Core Team, 2013*). Prior to statistical analysis, receptor density values were normalized by z-scores within each receptor type, and for all tests significance threshold was set at p≤0.05.

A first omnibus test of all regions and receptors was performed to establish if there were any significant differences in receptor density between all regions and receptor types, with a fixed effect for area. The random effects in the model consisted in a random intercept for individual subject and receptor type.

A second set of tests was used to determine if pairs of adjacent regions were significantly different from one another over all receptor types. The linear mixed effect model used for the second

series of tests had the same form as that of the omnibus test, but was only applied to pairs of adjacent regions. The p-values for the main effect 'area' were corrected for multiple comparisons using the Benjamini-Hochberg correction for false-discovery rate (*Benjamini and Hochberg, 1995*).

Finally, for pairs of areas that were significantly different from one another in the second level tests after correction for multiple comparisons, we ran separate linear mixed-effect models for each receptor type to test for a difference in the density between paired regions. The model was composed of a fixed effect for area and a random intercept for each macaque brain. The p-values for the fixed-effect 'area' from each of these tests (equals number of pairs of areas multiplied by the 13 of receptor types) were again corrected using the Benjamini-Hochberg correction for false-discovery rate (*Benjamini and Hochberg, 1995*).

Subsequently, receptor density values were averaged over the four hemispheres, providing the mean value and s.d. for each receptor in each area. The mean density of each of the 15 receptor types in a given area was registered to a polar plot. The resulting graph constitutes the receptor fingerprint (*Zilles et al., 2002a*) of each area. Receptor fingerprints were treated as feature vectors characterizing the actual cortical area to show the multi-receptor balance in each subdivision of the IPS, and were the basis for subsequent hierarchical cluster analyses (MATLAB, Statistics Toolbox, The MathWorks Inc). For the cluster analyses, receptor densities were normalized by z-scores to ensure that all receptor types had equal weight, since they differed considerably in their absolute densities. Hierarchical cluster analyses were applied to determine grouping of areas based on similarities in their receptor fingerprints. A subsequent k-means analysis was applied to determine the acceptable number of clusters. Euclidean distances were computed as measures of the similarities between receptor fingerprints of IPS subdivisions. Thus, the smaller the Euclidean distance between two areas, the greater the similarity in shape and size of their fingerprints. A principal component analysis (PCA) of the fingerprints of all cortical areas of the IPS was performed to reduce the 15-dimensional space (15 different receptor types) into two dimensions for graphical representation of the Euclidean distances between areas.

## Acknowledgements

This project has received funding from the European Union's Horizon 2020 Research and Innovation Programme under the Specific Grant Agreements 785907 (Human Brain Project SGA2) and 945539 (Human Brain Project SGA3), and from the BMBF (01GQ1902).

## Additional information

### Funding

| Funder | Grant reference number | Author |
| --- | --- | --- |
| European Commission | 785907 | Nicola Palomero-Gallagher Karl Zilles |
| Bundesministerium für Bildung und Forschung | 01GQ1902 | Nicola Palomero-Gallagher |
| European Commission | 945539 | Nicola Palomero-Gallagher Karl Zilles |

The funders had no role in study design, data collection and interpretation, or the decision to submit the work for publication.

### Author contributions

Meiqi Niu, Conceptualization, Formal analysis, Investigation, Visualization, Writing - original draft, Writing - review and editing, Border identification; Daniele Impieri, Lucija Rapan, Border identification; Thomas Funck, Formal analysis; Nicola Palomero-Gallagher, Conceptualization, Resources, Data curation, Supervision, Funding acquisition, Methodology, Project administration, Writing - review and editing, Border identification; Karl Zilles, Conceptualization, Resources, Formal analysis, Supervision, Funding acquisition, Methodology, Project administration, Writing - review and editing, Border identification

## Author ORCIDs

Meiqi Niu (iD) https://orcid.org/0000-0001-7937-5814
Nicola Palomero-Gallagher (iD) https://orcid.org/0000-0003-4463-8578

## Ethics

Animal experimentation: The present study did not include experimental procedures with live animals. Brains were obtained when animals were sacrificed to reduce the size of the colony, where they were maintained in accordance with the guidelines of the Directive 2010/63/eu of the European Parliament and of the Council on the protection of animals used for scientific purposes.

## Decision letter and Author response

Decision letter https://doi.org/10.7554/eLife.55979.sa1
Author response https://doi.org/10.7554/eLife.55979.sa2

## Additional files

### Supplementary files

• Transparent reporting form

### Data availability

All data generated or analysed during this study are included in the manuscript and supporting files.

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
