## [Decision Letter]

**Acceptance summary:**

Palomero-Gallagher report an exhaustive receptor density and cyto/myeloarchitectonic mapping study of the intraparietal sulcus of macaques. It is a remarkable dataset that give unique insights into anatomical divisions in the areas. It will be of high value to researchers who study parietal cortex function in humans as well as macaques.

**Decision letter after peer review:**

Thank you for submitting your article "Receptor-driven, multimodal mapping of cortical areas in the intraparietal sulcus of macaque monkey" for consideration by *eLife*. Your article has been reviewed by two peer reviewers, and the evaluation has been overseen by Timothy Behrens as the Senior Editor and Reviewing Editor. The following individual involved in review of your submission has agreed to reveal their identity: Wim Vanduffel (Reviewer #2).

The reviewers have discussed the reviews with one another and the Reviewing Editor has drafted this decision to help you prepare a revised submission.

Summary:

Niu and colleagues performed an exhaustive receptor density and cyto/myeloarchitectonic mapping study of the intraparietal sulcus of macaques. They confirmed the existence of multiple previously identified areas and were able to further subdivide some areas, arriving at a total of 17 areas occupying the IPS in monkeys. This reflects an incredibly rich data set which will be highly useful for anyone studying the parietal cortex of monkeys and humans. The authors used established methods from their own laboratory to acquire and analyze the data – an approach they used in many of their previous studies. The paper is well written and despite the lengthy discussion, it remains a digestible manuscript. From an empirical point of view, this study is highly commendable, yet the interpretation of the results in terms of anatomical-functional coupling of these different IPS areas might change in the future.

Revisions:

We found very little to fault with this manuscript. Here you can find some suggestions that we believe would be useful to clarify or discuss. Note that we do not require any changes except to the text.

1) Almost by definition, the author's starting points were existing maps of the IPS. Although many parcellation schemes exist, they mainly used the maps described in Lewis and Van Essen, 2000, which is a highly commendable landmark paper. One could wonder, however, if one would have started with another parcellation, or even if no a priori knowledge existed about the IPS areas, one would end up with the same 17 areas as defined here. The reason why I ask this is because some of the features distinguishing the different neighboring areas are exceedingly subtle (e.g. the differences in cell size and densities across the different layers of MIPd and MIPv, Figure 5), probably only discernible by very few expert anatomists. Also, the quantitative analyses (Mahalanobis distances, which relies on differences of vectorizes features across the cortical ribbon) show clear local minima and maxima, which are non-trivial to separate from the significant maxima). I assume that existing parcellations are used to determine whether it is a local maximum/minimum rather than a global one?

2) A related issue concerns the rather qualitative description of the results. Almost everywhere in the result section, one indicates that a particular receptor distribution is larger or smaller in area X versus area Y, without (much) statistical backup. Fingerprints are provided (Figure 11), and I would like to advice to test statistically for interareal differences (not just in a post hoc manner) taking into consideration all areas and receptors and correcting for multiple comparisons – in addition to the cluster analysis (Figure 12).

3) I think most of the areal boundaries and descriptions are defendable, although “LOP" is not frequently used anymore. Alternative descriptions for that part of cortex do exist. Also, the boundaries of area V3d and V3A are open for discussion – some authors (e.g. Nakamura et al., 2001 and Ungerleider et al., 2008) label much more of the annectant gyrus as area V3A. Also, the recent papers by Hadjidimitrakis et al., 2019 and Gamberini et al., 2019, provide a slightly alternative view on the layout of the posterior parts of the IPS (and SPL). This should be discussed.

4) Despite the exceedingly nice and useful data, this reviewer is not entirely convinced that one can easily draw conclusions concerning different functional streams just based on the receptor density data. It should be made clear (in Abstract and Discussion) that these conjectures are hypothetical.

5) This is a suggestion for clarification of the figures that you can choose to follow but feel free to ignore:

The presentation of the results is sometimes a bit confusing for a reader who has not spend several months looking at the data in great detail. For instance, the receptor fingerprints (Figure 11) show very subtle differences. If they authors want the reader to appreciate the gradients of change that are present in these data, a representation where the fingerprints are ordered along the 2D anatomical maps of the other figures, this might be more apparent.

---

## [Author Response]

Revisions:[…]1) Almost by definition, the author's starting points were existing maps of the IPS. Although many parcellation schemes exist, they mainly used the maps described in Lewis and Van Essen, 2000, which is a highly commendable landmark paper. One could wonder, however, if one would have started with another parcellation, or even if no a priori knowledge existed about the IPS areas, one would end up with the same 17 areas as defined here. The reason why I ask this is because some of the features distinguishing the different neighboring areas are exceedingly subtle (e.g. the differences in cell size and densities across the different layers of MIPd and MIPv, Figure 5), probably only discernible by very few expert anatomists. Also, the quantitative analyses (Mahalanobis distances, which relies on differences of vectorizes features across the cortical ribbon) show clear local minima and maxima, which are non-trivial to separate from the 'significant maxima). I assume that existing parcellations are used to determine whether it is a local maximum/minimum rather than a global one?

We agree with the reviewer that this is a very important question, which should be clarified. The reviewer is correct in the assumption that we used existing maps of the IPS as a starting point, though not in the one that we mainly used that by Lewis and Van Essen. Rather, it was the result of our quantitative analysis that proved theirs to be the most precise of the previously published parcellation schemes. Furthermore, the diversity of previous maps, which were based solely on visual inspection, is not only due to the subtlety of interareal differences in cytoarchitecture, but also to the existence of even more subtle variations within an area, which are reflected in the local minima and maxima of our Mahalanobis distance function. As the reviewer said, we calculated the Mahalanobis distances at each position, and a maximum of the Mahalanobis distance function indicated potential architectonical border. However, it was only accepted as such when the p-value of the subsequent Hotelling’s T2 test was significant after Bonferroni correction. Furthermore, since the presence of blood vessels can at times influence the Mahalanobis distance, the biological relevance of significant maxima was confirmed by microscopic inspection of the histological section and by comparing spatially corresponding significant maxima from adjoining sections. Since the probability of locating such maxima at comparable sites in a stack of sections by reasons other than changes in cortical organization is exceedingly small, this procedure excluded biologically meaningless maxima which may be caused by artifacts (e.g., ruptures, folds) or local inhomogeneities in microstructure such as blood vessels or atypical cell clusters. These latter points have now been included in the Material and Methods section.

2) A related issue concerns the rather qualitative description of the results. Almost everywhere in the result section, one indicates that a particular receptor distribution is larger or smaller in area X versus area Y, without (much) statistical backup. Fingerprints are provided (Figure 11), and I would like to advice to test statistically for interareal differences (not just in a post hoc manner) taking into consideration all areas and receptors and correcting for multiple comparisons – in addition to the cluster analysis (Figure 12).

Following this valuable suggestion, we carried out a statistical analysis to identify significant differences in receptor densities between adjacent regions. The following paragraphs have been included in the Materials and methods section:

“In order to identify significant differences in receptor densities between adjacent cortical areas, we performed a series of linear mixed-effects models using the R programming language (version: 3.6.3; R Development Core Team, 2013). […] The p-values for the fixed-effect “area” from each of these tests (equals number of pairs of areas multiplied by the 13 of receptor types) were again corrected using the Benjamini-Hochberg correction for false-discovery rate.”

We have also updated the Results section to include a description of statistically significant differences in densities between pairs of neighboring areas.

Notably, not all receptors show each border in the equally clear way and each receptor cannot indicate all areal borders. Nevertheless, if a border can be detected by several receptor types, and this happened at a comparable position in a series neighboring sections, we confirmed the existence of the identified border.

Additionally, receptor architectonic variations represented not only in the mean absolute densities (averaged over all cortical layers), but also in particularly laminar variations. The data of laminar-specific analysis will not be presented in this article. We plan to use it for a more detailed investigation in the future.

3) I think most of the areal boundaries and descriptions are defendable, although “LOP" is not frequently used anymore. Alternative descriptions for that part of cortex do exist. Also, the boundaries of area V3d and V3A are open for discussion – some authors (e.g. Nakamura et al., 2001 and Ungerleider et al., 2008) label much more of the annectant gyrus as area V3A. Also, the recent papers by Hadjidimitrakis et al., 2019 and Gamberini et al., 2019, provide a slightly alternative view on the layout of the posterior parts of the IPS (and SPL). This should be discussed.

We appreciate for the valuable comments. Following your suggestions, we have updated the Discussion in the revised manuscript and created an additional accompanying figure to specifically address the issue of controversies concerning the border between areas V3d and V3A.

“Most parcellation schemes agree on the existence of areas V3A, V3d and PIP in the fundus of the IPS/POS junction, and we were able to confirm these observations. […] In general, the present study provides evidence to confirm that LOP/CIP could be designated unequivocally as a distinct area.”

4) Despite the exceedingly nice and useful data, this reviewer is not entirely convinced that one can easily draw conclusions concerning different functional streams just based on the receptor density data. It should be made clear (in Abstract and Discussion) that these conjectures are hypothetical.

We agree with the reviewer that it is not possible to draw conclusions concerning different functional streams based solely on differences in receptor densities and have modified both the Abstract and Discussion accordingly.

Abstract:

“The intraparietal sulcus (IPS) is structurally and functionally heterogeneous. […] Thus, differences in cyto- and receptor architecture segregate the cortical ribbon within the IPS, and receptor fingerprints provide novel insights into the relationship between the structural and functional segregation of this brain region in the macaque monkey.”

Discussion:

“One of the purposes of this study was to analyze the organization of the IPS with a focus on the relationship between its molecular and functional segregation since similarities in fingerprints have been postulated to be indicative of the existence of a network of cortical areas characterized by their comparable receptor expression and participation in the same functional system. Furthermore, since the function of a cortical area requires a well-tuned receptor balance, the receptor fingerprint of each area may represent an index of its physiological, connectional and functional properties.”

Discussion:

“Interestingly, this shift in functional systems, from a caudo-lateral visual domain to a rostro-medial sensorimotor domain, is also the most significant trend reflected by changes in the shape and/or size of the fingerprints.”

5) This is a suggestion for clarification of the figures that you can choose to follow but feel free to ignore:The presentation of the results is sometimes a bit confusing for a reader who has not spend several months looking at the data in great detail. For instance, the receptor fingerprints (Figure 11) show very subtle differences. If they authors want the reader to appreciate the gradients of change that are present in these data, a representation where the fingerprints are ordered along the 2D anatomical maps of the other figures, this might be more apparent.

We appreciate the reviewer’s suggestion. It is true that the shape and size of receptor fingerprints are dominated by the receptor types present at highest absolute densities in the brain (e.g. GABA_A_). Thus, area-specific differences in the densities of receptors expressed at lower levels in the brain (e.g., 5-HT_2_) are not always obvious. We have created a supplementary figure (Figure 11—figure supplement 1) presenting normalized receptor fingerprints, which better reflect differences in shape than do the fingerprints of absolute densities, although differences in size are not as obvious.